# Phenethyl Isothiocyanate-Conjugated Chitosan Oligosaccharide Nanophotosensitizers for Photodynamic Treatment of Human Cancer Cells

**DOI:** 10.3390/ijms232213802

**Published:** 2022-11-09

**Authors:** Inho Bae, Taeyu Grace Kim, Taeyeon Kim, Dohoon Kim, Doug-Hoon Kim, Jaewon Jo, Young-Ju Lee, Young-Il Jeong

**Affiliations:** 1Department of Dental Materials, College of Dentistry, Chosun University, Gwangju 61452, Korea; 2Tyros Biotechnology Inc., 75 Kneeland St. 14 floors, Boston, MA 02111, USA; 3Brookline High School, 115 Greenough St., Brookline, MA 02445, USA; 4College of Art & Science, University of Pennsylvania, 249 S 36th St., Philadelphia, PA 19104, USA; 5Department of Optometry, Masan University, Changwon 51217, Korea; 6Gwangju Center, Korea Basic Science Institute, Gwangju 61186, Korea

**Keywords:** photodynamic therapy, oral cancers, nanophotosensitizers, reactive oxygen species, tumor-targeting

## Abstract

The aim of this study is to synthesize phenethyl-conjugated chitosan oligosaccharide (COS) (abbreviated as ChitoPEITC) conjugates and then fabricate chlorin E6 (Ce6)-incorporated nanophotosensitizers for photodynamic therapy (PDT) of HCT-116 colon carcinoma cells. PEITC was conjugated with the amine group of COS. Ce6-incorporated nanophotosensitizers using ChitoPEITC (ChitoPEITC nanophotosensitizers) were fabricated by dialysis method. ^1^H nuclear magnetic resonance (NMR) spectra showed that specific peaks of COS and PEITC were observed at ChitoPEITC conjugates. Transmission electron microscope (TEM) confirmed that ChitoPEITC nanophotosensitizers have spherical shapes with small hydrodynamic diameters less than 200 nm. The higher PEITC contents in the ChitoPEITC copolymer resulted in a slower release rate of Ce6 from nanophotosensitizers. Furthermore, the higher Ce6 contents resulted in a slower release rate of Ce6. In cell culture study, ChitoPEITC nanophotosensitizers showed low toxicity against normal CCD986Sk human skin fibroblast cells and HCT-116 human colon carcinoma cells in the absence of light irradiation. ChitoPEITC nanophotosensitizers showed a significantly higher Ce6 uptake ratio than that of free Ce6. Under light irradiation, cellular reactive oxygen species (ROS) production of nanophotosensitizers was significantly higher than that of free Ce6. Especially, PEITC and/or ChitoPEITC themselves contributed to the production of cellular ROS regardless of light irradiation. ChitoPEITC nanophotosensitizers showed significantly higher PDT efficacy against HCT-116 cells than that of free Ce6. These results indicate that ChitoPEITC nanophotosensitizers have superior potential in Ce6 uptake, ROS production and PDT efficacy. In the HCT-116 cell-bearing mice tumor-xenograft model, ChitoPEITC nanophotosensitizers efficiently inhibited growth of tumor volume rather than free Ce6. In the animal imaging study, ChitoPEITC nanophotosensitizers were concentrated in the tumor tissue, i.e., fluorescence intensity in the tumor tissue was stronger than that of other tissues. We suggest that ChitoPEITC nanophotosensitizers are a promising candidate for the treatment of human colon cancer cells.

## 1. Introduction

Photodynamic therapy (PDT) is regarded as an ideal treatment option for inoperable cancer because it uses light, oxygen and photosensitizers [1,2]. PDT has minor side effects compared to chemotherapy or radiotherapy since it produces reactive oxygen species (ROS) under light irradiation, i.e., ROS can be produced in the specific site of light irradiation [3,4]. These intrinsic properties of PDT enable us to minimize undesirable medicinal effects in normal organs since photosensitizers including chlorin e6 (Ce6) produce ROS excessively in tumor tissues and then kill cancer cells [4,5,6]. However, traditional photosensitizers such as 5-aminolevulinic acid (5-ALA), porphyrin derivatives and chlorin e6 (Ce6) have several disadvantages for clinical use [6,7,8,9]. For example, PDT with traditional photosensitizers requires long-term sunshade because photosensitizers have limited tumor-delivering capacity and low cancer cell specificity, and they spread out in the whole body through systemic administration [7,8,9,10]. Thus, novel photosensitizers should be developed for efficient and specific delivery of photosensitizers for cancer treatment.

ROS in PDT is important in killing cancer cells [3,11]. Photosensitizers produce excessive ROS locally by irradiation of visible light in tumor tissues, and then, they induce apoptosis/necrosis of cells even though they are simultaneously distributed in normal cells/tissues [12]. Otherwise, ROS is known as a as double-edged sword, i.e., appropriate levels of ROS have a critical role in the maintenance of homeostasis of cells, and they are involved in the mitosis, migration or proliferation/differentiation of normal cells [12,13,14]. These critical roles of ROS produced by photosensitizers can be applicable to cancer therapy through site-specific overproduction in tumor tissues. Therefore, therapeutic potential of PDT in tumor tissues is principally dependent on the efficacy of ROS production by photosensitizers [14,15]. When cells are under oxidative stress induced by PDT, a defensive mechanism is activated to regulate intracellular ROS levels and to maintain homeostasis of cells, i.e., intracellular ROS level is regulated by intracellular antioxidant molecules such as reduced glutathione (GSH) or heme oxygenase-1 (HO-1) [16,17,18]. For example, Lee et al. reported that intracellular glutathione inhibits the accumulation of intracellular ROS in PDT of cholangiocarcinoma cells, while Ce6-based PDT dose-dependently decreases the total GSH level in cells [16]. Musaogullari et al. reported that accumulation of intracellular ROS can be enhanced by inhibition of antioxidant molecules [17]. Therefore, ROS generation and accumulation in the intracellular compartment is important to the success of PDT in cancer cells. Especially, Jia et al. reported that phenethyl isothiocyanate (PEITC) can be used as a ROS-inducing agent since PEITC inhibits ROS scavengers and then synergistically enhances the cytotoxic effects of BMN 673 through overproduction of ROS, inducing DNA damage and apoptosis [19]. Liu et al. also reported that PEITC efficiently depletes L-glutathione in the cells, which is a ROS scavenger, and then over-production of ROS follows with synergistic death of cancer cells [20].

Nano-scale carriers such as nanoparticles and polymeric micelles have been extensively used to improve the delivery of photosensitizers, production of ROS and the efficacy of PDT [21,22,23,24]. Since nanoparticles have small hydrodynamic diameters, they are ideal candidates for tumor drug targeting through the avoidance of reticuloendothelial system (RES) accumulation and enhanced uptake by tumor tissues [21]. Furthermore, they can be easily modulated to be sensitive against the tumor microenvironment, and they specifically liberate anticancer drugs [21,22,23]. Mehraban et al. reported that polymeric micelles or nanoparticles as a drug delivery system can be used to improve intracellular delivery of photosensitizers and then produce/accumulate intracellular ROS [24]. There are many reports regarding nanoparticles for the assistance of ROS production [25,26,27]. For example, Zhao et al. reported that sonosensitizer-loaded nanoparticles generate ROS and that they can be used for therapeutic or diagnostic purposes [26]. Glass et al. reported that highly tunable nanoparticle-based drug delivery systems improve the therapeutic potential of biological agents, i.e., nanoparticles-assisted chemotherapy through redox-sensitive drug-release in tumor cells, PDT by enhancing ROS production, and radiation therapy by ROS production [25]. Most of the nanoparticulate drug-delivery systems provide a delivery platform for anticancer drugs or photosensitizers and assist in ROS production.

In this study, we synthesized phenethyl isothiocyanate (PEITC)-conjugated chitosan oligosaccharide (COS) (abbreviated as ChitoPEITC) for the enhancement of Ce6 cellular delivery and cellular ROS production in cancer cells. ChitoPEITC nanophotosensitizers were fabricated to study the effect of ChitoPEITC on the generation of ROS and PDT against HCT-116 human colon carcinoma cells. Physicochemical and biological characters were investigated in vitro and in vivo.

## 2. Results

### 2.1. Synthesis and Characterization of ChitoPEITC Conjugates

^1^H and ^13^C nuclear magnetic resonance (NMR) spectra of COS show specific peaks of H1~H7 as shown in Appendix A. That is, the acetyl group of COS was observed at 1.85 ppm, and H3~H6 was observed at 3.4~3.9 ppm. H1 and H2 were observed at 4.6~4.8 and 2.7~3.0 ppm. Furthermore, ^13^C NMR of COS showed specific peaks at each position of carbon as shown in Appendix A. To calculate the deacetylation degree of COS, an integral value between H3~H6 and H7 was compared and calculated, and the deacetylation degree was approximately 96% as shown in Appendix A. Appendix A shows the quantitative NMR (qNMR) of COS for the evaluation of the molecular weight (M.W.) of COS. For the evaluation of M.W. of COS, dimethylmalonic acid was used as a standard material. The integral value of H2 and the methyl proton of dimethylmalonic acid were compared, and then, the typical M.W. of COS was calculated as 1150 g/mol. Furthermore, the analysis of COS using matrix-assisted laser desorption ionization mass (MALDI TOF/TOF MS) spectrometer showed that COS was composed of various M.W. of oligosaccharides as shown in Appendix A.

Since PEITC has an isothiocyanate group in its intrinsic structure, they can be easily conjugated with the amine group of COS, of which the byproducts are removed by dialysis procedure. As shown in Figure 1a,b, COS and PEITC were simply mixed and reacted at room temperature. COS has intrinsic proton peaks at 1.5~5.0 ppm, and PEITC has specific peaks at 2.0~8.0 ppm. When PEITC was conjugated with COS, specific peaks of COS and PEITC were observed between 1.5 and 8.0 as shown in Figure 1b. Substitution degree (DS) of PEITC in 100 glucosamine was estimated from proton peaks of COS and PEITC as shown in Table 1. ChitoPEITC conjugates are able to form nanoparticles because PEITC has hydrophobic properties and COS is a water-soluble molecule. These properties must be induced aggregates as nano-sized vehicles in the aqueous solution. Therefore, hydrodynamic diameter was measured whether or not ChitoPEITC conjugates can form nanoparticles, as shown in Table 1. When DS of PEITC was increased, hydrodynamic diameters were gradually increased.

### 2.2. Characterization of Ce6-Incorporated ChitoPEITC Nanophotosensitizers

To make Ce6-incorporated nanoparticles, ChitoPEITC solution was mixed with Ce6 solution, and then organic solvent or free drug was removed from Ce6-incorporated PEITC nanoparticles (ChitoPEITC nanophotosensitizers). To confirm formation of nanophotosensitizers, their morphology was observed using TEM, and their hydrodynamic diameters were measured as shown in Figure 2. As shown in Figure 2a,b, ChitoPEITC nanophotosensitizers have spherical shapes with small diameters less than 200 nm, and their hydrodynamic diameter distributions were 169 ± 29 nm. These results indicated that ChitoPEITC nanoparticles formed small nano-dimensional particles and have small diameters, which is acceptable for intravenous administration to animals. Table 2 summarizes the characterization of empty ChitoPEITC nanophotosensitizers. As shown in Table 2, a higher feeding ratio of Ce6 resulted in higher drug contents and showed increased hydrodynamic diameters. Furthermore, drug contents were slightly increased at higher DS of PEITC. These results might be because hydrophobic interactions between Ce6 and PEITC change the drug contents, thus increasing the hydrodynamic diameter.

To analyze the ultraviolet–visible (UV) spectra of nanophotosensitizers, free Ce6, Ce6-incorporated nanophotosensitizers and empty nanoparticles were measured with a UV–VIS spectrophotometer. As shown in Figure 3, free Ce6 showed specific a peak intensity at 200~800 nm, and maximum peaks were observed at 664 nm. Ce6-incorporated nanophotosensitizers show an almost similar peak intensity at 300~800 nm, while no specific peak intensity was observed in the field with empty nanoparticles. These results indicate that no chemical modification of Ce6 itself occurred during the fabrication process of the nanophotosensitizers.

To analyze the stability of the hydrodynamic diameter of nanoparticles, Ce6-incorporated ChitoPEITC nanoparticles (ChitoPEITC NP) in deionized water were mixed with deionized water, phosphate buffered saline and Roswell Park Memorial Institute (RPMI1)-1640 media.

As shown in Figure 4a–c, the average hydrodynamic diameter was slightly increased according to time course in all aqueous solutions. Practically, some of precipitants were observed in the deionized water and PBS of ChitoPEITC-2 and 3 NP sample after 10 days, even though they can be easily reconstituted in the media. Even though average hydrodynamic diameter was slightly increased, the stability of ChitoPEITC NP was properly maintained until 7 days in deionized water (Figure 4a), PBS (Figure 4b) and RPMI-1640 media (Figure 4c), Especially, their average hydrodynamic diameter was slightly higher in the PBS or PRMI-1640 media than of those in deionized water.

Ce6 release study was performed in vitro as shown in Figure 5. Figure 5 shows that Ce6 was continuously released from the nanophotosensitizers. Ce6 release rate from Ce6-incorporated nanophotosensitizers was almost a continuous pattern until 48 h, and then, Ce6 was slightly slowly released until 96 h. As shown in Figure 5a, the higher PEITC ratio in the ChitoPEITC conjugates resulted in a slower release from the nanophotosensitizers. Furthermore, the higher Ce6 contents in the nanophotosensitizers resulted in a slower release rate from the nanophotosensitizers (Figure 5b). These results might be due to the hydrophobic interaction between Ce6 and PEITC. Ce6 was continuously released from the nanoparticles over 4 days for all nanophotosensitizers.

### 2.3. In Vitro Cell Culture Study

Intrinsic cytotoxicity as a dark toxicity was evaluated with CCD986Sk human skin fibroblast cells and HCT-116 human colon cancer cells as shown in Figure 6. As shown in Figure 6a, free Ce6 or ChitoPEITC nanophotosensitizers showed low cytotoxicity in the absence of light irradiation against CCD986Sk cells. Furthermore, free Ce6 or ChitoPEITC nanophotosensitizers also showed low cytotoxicity in the absence of light irradiation against HCT-116 cells as shown in Figure 6b. Viability of CCD986Sk cells was slightly decreased until 2 μg/mL of Ce6 concentration (Figure 6a). HCT-116 cell viability was also slightly decreased at concentrations higher than 0.5 μg/mL. However, the viability of cells was higher than 80% at all samples. These results indicated that ChitoPEITC nanophotosensitizers have little toxicity against normal cells (CCD986Sk cells) and cancer cells (HCT-116 cells). Furthermore, these results indicated that nanophotosensitizers do not significantly inhibit the viability of tumor cells in the absence of light as well as free Ce6.

Figure 7 shows the cellular Ce6 uptake ratio of cancer cells. As shown in Figure 7a, cellular Ce6 uptake ratio was gradually increased according to the Ce6 concentration of all samples. Cellular Ce6 levels were relatively higher in higher PEITC contents of ChitoPEITC copolymers. Especially, nanophotosensitizers showed significantly higher cellular Ce6 uptake compared to free Ce6 as shown in Figure 7a. Furthermore, red fluorescence color of cancer cells was significantly higher in the ChitoPEITC-2 NP than free Ce6, indicating that the cellular Ce6 uptake of nanophotosensitizers was significantly higher than that of free Ce6. These results indicated that nanophotosensitizers accelerate cellular delivery of Ce6 and/or can be internalized into the cellular compartment.

Figure 8 shows the cellular ROS generation after treatment of free Ce6 or nanophotosensitizers. When free Ce6 or nanophotosensitizers were treated to HCT-116 cells, cellular ROS level was significantly increased by light irradiation at 2 J/cm^2^, while ROS level was not significantly increased at 0 J/cm^2^ as shown in Figure 8a,b. Cellular ROS level was slightly increased according to the concentration of Ce6 in the absence of light irradiation as shown in Figure 8a. Especially, the higher PEITC contents in the ChitoPEITC copolymer induced a slightly higher cellular ROS level. Then, we measured ROS level with PEITC or ChitoPEITC conjugates as shown in Figure 8c,d. As shown in Figure 8c, cellular ROS level was gradually increased according to the concentration of PEITC or ChitoPEITC copolymer in the absence of light irradiation. Furthermore, cellular ROS level was also increased under light irradiation (Figure 8d), but the ROS level was not significantly changed compared to the absence of light irradiation. These results indicate that ChitoPEITC conjugates may synergistically contribute to ROS generation as well as PEITC itself. These results indicate that ChitoPEITC copolymers themselves can significantly contribute to cellular ROS generation.

PDT efficacy of nanophotosensitizers was performed in vitro. As shown in Figure 9a, viability of HCT-116 cells was dose-dependently decreased at higher than 0.1 μg/ ml Ce6 concentration. The viability of HCT-116 cells was gradually decreased when free Ce6 or nanophotosensitizers were treated to cells and then irradiated at 664 nm, as shown in Figure 9a. Nanophotosensitizers showed excellent PDT efficacy, i.e., cell viability was less than 30% at 2 μg/mL Ce6 concentration while free Ce6 had lower PDT efficacy compared to nanophotosensitizers. Even though PEITC or ChitoPEITC nanophotosensitizers showed negligible PDT efficacy (Figure 9b), the higher PEITC contents in the ChitoPEITC conjugates resulted in a lower viability of HCT-116 cells, i.e., cell viability was decreased according to PEITC or ChitoPEITC concentration at higher than 10 μg/ mL. These results might be because PEITC may contribute to the production of ROS, thus inducing death of cancer cells. Therefore, PEITC may contribute to synergistic anticancer effects with nanophotosensitizers.

### 2.4. Animal Tumor Imaging of Tumor Xenograft Model

PDT efficacy of nanophotosensitizers was investigated with HCT-116-bearing mice as shown in Figure 10. As shown in Figure 10a, tumor volume was gradually increased with time course. When free Ce6 or nanophotosensitizers were injected with light irradiation, the tumor volume was relatively smaller than those of control or empty nanoparticles at 30 days. Practically, empty nanoparticles did not affect changes in tumor volume. Among all treatments, nanophotosensitizers properly delayed the changes in tumor volume compared to free Ce6 and empty nanoparticles as shown in Figure 10a. Figure 10b supports these results, i.e., strong fluorescence intensity was observed in tumor xenograft at whole body imaging of the mice. Furthermore, fluorescence images of organs showed that fluorescence intensity was strongest in the tumor tissues compared to other organs. These results indicated that ChitoPEITC nanophotosensitizers can be efficiently accumulated in the tumor tissue and induce superior PDT efficacy.

## 3. Discussion

Physiological features of the tumor microenvironment are quite different from normal tissues or cells [28,29,30]. Abnormal physicochemical status can be expressed as acidic pH, increased reduction/oxidation (redox) potential, overexpression of various protein/receptors and elevated metabolism [28,29,30]. Amon them, elevated redox potential is known to be increased in the tumor microenvironment due to increased metabolism [28]. Cellular ROS level is known to be elevated in tumor tissues/cancer cells, of which this status makes it difficult to cure cancer. Paradoxically, L-glutathione (GSH), a typical strong antioxidant molecule, is also increased in the tumor tissues or cancer cells to maintain homeostasis of the tumor and as a defensive mechanism against oxidative stress [29]. Increased GSH level in tumors has a critical role in the therapeutic strategy of cancer patients, i.e., high GSH level in tumors leads to drug-resistant problems in the tumor and then disturbs the therapeutic efficacy of chemotherapy/PDT [31,32]. Since therapeutic efficacy of PDT is primarily dominated by the accumulation of cellular ROS level, PDT efficacy for tumor treatment is frequently deteriorated by cellular GSH level [16]. Therefore, ROS production may be a critical issue to treat cancer patients with PDT. Lee et al. reported that PDT efficacy can be improved through efficient cellular ROS level by inhibition of cellular GSH level [33]. Cellular ROS accumulation can be altered by use of ROS-producing chemical agents. For example, Guo et al. reported that artesunate, a ROS-producing agent, synergistically increased ROS production and then induced effective inhibition of the proliferation of cancer cells [34]. Hu et al. reported that ROS production and PDT efficacy of indocyanine-loaded nanoparticles can be improved by co-loading with PEITC [35]. They reported that co-loading of PEITC into indocyanine-loaded nanoparticles contributed to producing ROS by depletion of GSH, which then synergistically suppressed the viability of cancer cells. Our results also showed that PEITC generates cellular ROS in cancer cells in a dose-dependent manner, and empty ChitoPEITC nanoparticles also produce cellular ROS similar to PEITC itself (Figure 8c,d). When nanoparticles are internalized in the cells, they can be partly dissociated due to the acidic pH of the endosome and endosomal enzymes following with cellular digestion of the nanoparticles [36]. Due to these properties in the cellular fate of nanoparticles, drug release must be accelerated inside the cell. Furthermore, ChitoPEITC nanophotosensitizers showed higher Ce6 uptake and PDT efficacy compared to free Ce6 (Figure 7 and Figure 9). Especially, PEITC and ChitoPEITC-empty nanoparticles also show cytotoxicity at higher than 10 μg/mL, as shown in Figure 9b. These results indicate that ChitoPEITC-empty nanoparticles also synergistically contribute to ROS production as well as PEITC regardless of light irradiation. Even though ChitoPEITC nanophotosensitizers also showed little cytotoxicity at higher than 1 μg/mL and cell viability was slightly decreased (Figure 6), ChitoPEITC nanophotosensitizers efficiently concentrated in the tumor tissue, as shown in Figure 10b, where they efficiently inhibited growth in tumor volume. Other researchers also reported that PEITC synergistically suppressed cancer cells in vitro and in vivo [37,38,39]. Lv et al. also reported that the viability of K7M2 murine osteosarcoma cells can be inhibited by treatment of PEITC in a dose-dependent manner through ROS generation and apoptotic death [37]. Shoaib et al. reported that PEITC produces cellular ROS in a dose-dependent manner and then induces apoptotic death of cervical cancer cells [38]. PEITC is regarded as an effective tool for cellular ROS accumulation in cancer cells [39].

Primarily, cationic polymers such as chitosan or protein are regarded as ideal candidates for the incorporation of Ce6 because Ce6 has anionic three carboxylic acid [40,41]. Mojzisova et al. reported that Ce6 showed higher solubility in pH 7.4, and its solubility was significant in the acidic solution [40]. Jeong et al. reported that the cationic polymer, chitosan, forms ion-complexes with Ce6 by forming nanoparticles, and then enhances intracellular uptake in vitro [41]. Especially, nano-scale carriers such as nanoparticles are frequently considered as a controlled release and site-specific delivery of anticancer agents [41,42,43,44,45,46,47,48]. Nanoparticles can target tumor tissues through a targeting mechanism such as the enhanced permeation and retention (EPR) effect [44]. Nanoparticles can concentrate anticancer drugs in tumor tissue and then selectively kill cancer cells with minimal drug accumulation in normal tissues [43,44]. In our results, nanophotosensitizers effectively accumulated in the tumor tissues as shown in Figure 10b. These results might induce improved PDT efficacy of nanophotosensitizers as shown in Figure 9. Since low tumor specificity and nonspecific dispersion against normal cells or tissues of traditional photosensitizers are frequently raised as major drawbacks, tumor specificity of nanophotosensitizers can improve PDT efficacy through an increase in photosensitizer concentration in the tumor tissues with higher ROS accumulation [45,46,47]. Additionally, nanoparticles are regarded as a suitable carrier for the absorption of photosensitizers against epithelial cancer or squamous cancer cells, while free photosensitizers revealed a low uptake ratio against tumors and were immediately removed from the target site [46]. In this study, Jeong et al. reported that Ce6-chitosan complexes also improve Ce5 uptake/ROS generation/PDT efficacy against gastrointestinal (GI) cells, and furthermore, they efficiently target tumor tissues while free Ce6 rapidly disappeared. Since traditional photosensitizers are frequently dispersed in normal cells or tissues, PDT requires sunshade for a while to prevent skin irritation [47]. These adverse effects can be minimized through the specific delivery of photosensitizers using nanophotosensitizers to thus maximize PDT efficacy.

## 4. Materials and Methods

### 4.1. Chemicals

Chitosan oligosaccharide (COS) was purchased from Tokyo Chemical Industry (TCI) Co., Ltd. (Tokyo, Japan). From the manufacturer’s data, the origin of COS was crustacea, and deacetylation degree was approximately 94% (Figure 1b and Appendix A). Molecular weight was calculated as 1150 g/mol (Appendix A) from quantitative NMR (qNMR) spectra. Dimethyl sulfoxide (DMSO) was purchased from Tokyo Chemical Industry (TCI) Co., Ltd. (Tokyo, Japan). Chlorin e6 (Ce6) was obtained from Frontier Sci. Co. (Logan, UT, USA). Bicinchoninic acid (BCA) kit, phenethyl isothiocyanate (PEITC), phosphotungstic acid, 2′,7′-dichlorofluorescin diacetate (DCFH-DA), 3-(4,5-dimethyl2-thiazolyl)-2, 5-diphenyl-2H-tetrazolium bromide (MTT) and 2,2,2-tribromoethanol (avertin) were purchased from Sigma Aldrich Chem. Co. (St. Louis, MO, USA). Dialysis membranes with molecular weight cutoffs (MWCO) of 1000 Da were purchased from Spectrum Labs., Inc. (Rancho Dominguez, CA, USA).

### 4.2. Synthesis of ChitoPEITC Conjugates

ChitoPEITC conjugates were synthesized as follows: COS (360 mg, 2 mM as a glucosamine unit) was dissolved in 5 mL of 0.01 N HCl and then mixed with 10 mL of DMSO. PEITC (32.6 mg (0.2 mM), 65.2 (0.4 mM), 130.4 (0.8 mM)) was dissolved in 10 mL DMSO and added to COS solution. This mixture was stirred for 24 h at 20 °C, and after that, this solution was poured into 10 mL of deionized water. This solution was introduced into dialysis membranes to remove organic solvent and byproducts. During dialysis procedure (MWCO: 2000 Da), deionized water was exchanged at 2 h intervals for 12 h and then 6 h intervals for 36 h. Following this, dialyzed solution was analyzed or lyophilized for 2 days. Finally, yellowish solids were observed. The yield of ChitoPEITC was calculated as follows: Yield (%, *w*/*w*) = [(Feeding weight of PEITC + feeding weight of COS)/final weight of ChitoPEITC] × 100. Yield of ChitoPEITC was higher than 93% in all formulations.

### 4.3. Characterization of ChitoPEITC Conjugates

Chemical structure of conjugates was confirmed with ^1^H NMR spectra (500 mHz Agilent ProPulse NMR system, Agilent Tech. Inc., Santa Clara, CA, USA). Each component and conjugates were dissolved in mixtures of D_2_O/DMSO (1/1, *v*/*v* for COS. 0.2/0.8 for ChitoPEITC) for analysis.

### 4.4. Ce6-Incorporated Nanophotosensitizers of ChitoPEITC Conjugates

Ce6-incorporated nanophotosensitizers: ChitoPEITC (40 mg) was reconstituted in 5 mL of deionized water and mixed with 10 mL of DMSO. Ce6 (2 or 4 mg) was separately dissolved in 2 mL DMSO and then dropped into ChitoPEITC solution under magnetic stirring. This solution was sonicated using a bar-type ultra-sonicator for 60 s. This solution was magnetically stirred for 10 min and then dialyzed against water over 1 d. To avoid saturation of the drug, deionized water was exchanged every 2 h interval for 12 h and 3~4 h for 12 h. Resulting solution was used for drug release study, analysis or lyophilized for 2 days.

Ce6 contents in nanophotosensitizers: Volume of dialyzed solution was adjusted to 40 mL with deionized water. After that, 2 mL of dialyzed solution was mixed with 8 mL of DMSO. This mixture was diluted with DMSO more than 10 times. Ce6 concentration in this solution was measured with ultraviolet–visible (UV–VIS) spectrophotometer (Genesys 10 s UV–VIS spectrophotometer, Thermo Fisher Scientific, Waltham, MA, USA). Empty ChitoPEITC nanophotosensitizers were used for blank test. For comparison, 1 mg of Ce6 in DMSO (16 mL) was mixed with 20 mg of empty ChitoPEITC nanoparticles in 4 mL of water following with sonication using ultra-sonicator for 30 s (1 s × 30 times, Vibra-cell™, Sonics & Materials Inc., Newtown, CT, USA). This solution was diluted with DMSO more than 100 times and then compared with Ce6-incorportaed ChitoPEITC nanophotosensitizers.

Ce5 contents in the nanophotosensitizers were calculated with the following equation: Ce6 content (*w*/*w*, %) = (weight of Ce6/total weight of nanophotosensitizers) × 100. Loading efficiency (*w*/*w*, %) = (Ce6 weight in the nanophotosensitizers/feeding weight of Ce6) × 100.

To calculate Ce6 concentration, calibration curve was determined as follows [48]: 10 mg of Ce6 was dissolved in 10 mL of DMSO (1 mg/mL). This solution was diluted with deionized water/DMSO mixtures (1/9, *v*/*v*) 100 times. Genesys 10 s UV–VIS spectrophotometer (Thermo Fisher Scientific, Waltham, MA, USA) was employed to measure absorption spectra of this solution. Calibration curve was determined from 0.1 to 7 µg/mL of Ce6. Otherwise, Ce6 (1 mg/mL DMSO) was 30 times diluted with PBS for estimation of Ce6 concentration in the PBS, and then, calibration curve was determined in the range of 0.3~10 µg/mL of Ce6 concentration.

### 4.5. Characterization of Nanophotosensitizers

For morphological observation of nanophotosensitizers, transmission electron microscope (TEM, H7600, Hitachi Instruments Ltd., Tokyo, Japan) was employed. Nanophotosensitizer solution, prepared as described above (20 μL), was placed onto a carbon film-coated grid and, after that, was dried in the room temperature over 6 h. Phosphotungstic acid (0.1% *w*/*w* in deionized water) was used for negative staining of nanophotosensitizers. Nanophotosensitizers were observed at 80 kV.

Nano-Zetasizer (Nano-ZS, Malvern, Worcestershire, UK) was employed to analysis of hydrodynamic diameter. Concentration of nanophotosensitizers prepared as described above was adjusted to 0.1 mg/mL as ChitoPEITC weight with deionized water.

Ultraviolet–visible (UV–VIS) spectrophotometer (Genesys 10 s UV–VIS spectrophotometer, Thermo Fisher Scientific, Waltham, MA, USA) was employed to measure UV–VIS spectra of free Ce6 and nanophotosensitizers.

### 4.6. Drug Release Study

The volume of aqueous nanophotosensitizer solution prepared as described above was adjusted to 40 mL (1.0 mg/mL as a ChitoPEITC weight) with deionized water. This solution (5 mL) was introduced into dialysis membrane (MWCO: 2000 Da) and then immersed into 45 mL PBS (pH 7.4, 0.01 M) in conical tube. This was incubated at 37 °C under shaking at 100 rpm. PBS sampled at predetermined time intervals and whole PBS were replaced with fresh PBS. For comparison, pure Ce6 (0.24 mg), similar to Ce6-ChitoPEITC-2 (ChitoPEITC-2:Ce6 = 40:2 weight ratio in Table 2) was dissolved into 5 mL by ultrasonication for 30 s (1 s × 30 times, Vibra-cell™, Sonics & Materials Inc., Newtown, CT, USA). Then, this solution also introduced into dialysis membrane (MWCO: 2000 Da) and then immersed into 45 mL PBS (pH 7.4, 0.01 M) in conical tube. This was incubated at 37 °C under shaking at 100 rpm. Genesys 10 s UV–VIS spectrophotometer (Thermo Fisher Scientific, Waltham, MA, USA) was employed to measure absorption spectra of this solution. All experiments were carried out in dark condition, and the results were expressed as mean ± standard deviation (SD) from three different experiments.

### 4.7. Cell Culture Study

For cell culture study, HCT-116 human colon carcinoma cells and CCD986sk human skin fibroblast cells were purchased from the Korean Cell Line Bank Co. Ltd. (Seoul, Korea). HCT-116 cells and CCD986sk cells were maintained with RPMI 1640 media and IMDM media, respectively. Cell culture media were supplemented with 10% (*v*/*v*) fetal bovine serum and 1% (*v*/*v*) antibiotics. Cells were maintained in a 5% CO_2_ incubator at 37 °C.

### 4.8. PDT Treatment of Cancer Cells In Vitro

For PDT study, HCT-116 cells were seeded in 96 wells at a density of 2 × 10^4^ cells/well. Cells were cultured overnight and then washed with PBS to treat with Ce6 or nanophotosensitizers. For Ce6 treatment, Ce6 (10 mg) was dissolved in 10 mL DMSO (1 mg/mL) and then diluted with serum-free media more than 100 times. Final DMSO concentration was adjusted less than 0.5% (*v*/*v*). Aqueous nanophotosensitizer solution, prepared as described above, was filtered with a 1.2 µm syringe filter. Ce6 or nanophotosensitizers were diluted with serum-free media and then treated to cells. These were incubated for 2 h in a 5% CO_2_ incubator (37 °C). Following this, cells were washed with PBS twice, and then, serum-free media (100 µL) were added. Cells were irradiated at 664 nm using an expanded homogenous beam (SH Systems, Gwangju, Korea). Light dose for PDT was 2.0 J/cm^2^, and light intensity was measured with a photo radiometer (Delta Ohm, Padua, Italy). After PDT, cells were incubated for 24 h in CO_2_ (37 °C). Cell viability was measured with MTT cell proliferation assay. Then, 30 µL of MTT solution (5 mg/mL in PBS) was added to 96 wells and then further cultured for 4 h in a CO_2_ incubator. Following this, supernatants were removed and then replaced with 100 µL DMSO. Viability of cells was measured by absorbance at 570 nm using an Infinite M200 PRO microplate reader. Cell cultures and PDT procedure were carried out in dark conditions. For dark toxicity, cells were not irradiated.

### 4.9. Cellular Ce6 Uptake and ROS Generation of Oral Cancer Cells In Vitro

Cellular Ce6 uptake ratio was measured as follows: HCT-116 cells (2 × 10^4^ cells/well) were seeded in 96-well plate and cultured overnight. After that, cells were washed with PBS, and Ce6 or nanophotosensitizers in serum-free/phenol red-free media were treated to cells. Two hours later, media were discarded, and cells were washed with PBS twice. Then, cells were lysed with 50 µL of lysis buffer (GenDEPOT, Barker, TX, USA) to measure cellular Ce6 uptake ratio. Cellular Ce6 was measured with relative fluorescence intensity with an Infinite M200pro microplate reader (Tecan) (excitation wavelength: 407 nm, emission wavelength: 664 nm). Protein concentration in cell lysates was measured with BCA protein assay kit.

ROS generation was measured as follows: For measurement of ROS, DCFH-DA method was used. HCT-116 cells (2 × 10^4^ cells/well) were seeded in 96-well plate and cultured overnight. After that, cells were washed with PBS, and Ce6 or nanophotosensitizers in serum-free/phenol-red free media with DCFH-DA (final concentration: 20 µm) were treated to cells for 2 h. After that, cells were washed with PBS twice, and then, 100 µL of fresh serum-free/phenol red-free RPMI media was added. Cells were irradiated at 664 nm (2.0 J/cm^2^), and cellular ROS generation was analyzed at an excitation wavelength of 485 nm and emission wavelength of 535 nm using microplate reader (Infinite M200 PRO microplate reader (Tecan)).

Fluorescence observation: HCT-116 cells (2 × 10^5^) seeded on the cover glass in six-well plates were cultured overnight. These cells were washed with PBS and then treated with Ce6 or nanophotosensitizers. One hour later, cells were washed with PBS twice and then fixed with 4% paraformaldehyde (PFA) solution in PBS for 15 m. After that, cells were immobilized with immobilization solution (Immunomount, Thermo Electron Co. Pittsburgh, PA, USA). Observation of cells was carried out with a fluorescence microscope (Emission bandwidth: 600–660 nm) (Eclipes 80i; Nikon, Tokyo, Japan). For observation of Ce6, red filter was used and bright field also observed.

### 4.10. Animal Tumor Imaging of HCT-116-Bearing Tumor Xenograft Model

For imaging of tumor targeting of ChitoPEITC nanophotosensitizers, HCT-116 cells (1 × 10^6^ cells) were subcutaneously (s.c.) administered into the backs of nude BALb/C mice (male, 20 g, five weeks old, OrientBio Co. Ltd. Seongnam-si, Gyeonggido, Korea). When the diameter of the tumor xenograft became larger than 6 mm, HCT-116-cell-bearing mice were used for fluorescence imaging. ChitoPEITC nanophotosensitizers in aqueous solution were administered intravenously (i.v.) via the tail vein of the mice. Injection volume was 200 µL, and as a Ce6 dose, 10 mg/kg was used. MaestroTM 2 small animal imaging instrument (Cambridge Research and Instruments, Inc. Woburn, MA, USA) was employed to observe whole body imaging of the mouse. For fluorescence imaging of organs, mice were sacrificed and then observed.

PDT efficacy in vivo was carried out using HCT-116-bearing mice. Then, 1 × 10^6^ HCT-116 cells were administered subcutaneously (s.c.) into the backs of nude BALb/C mice (20 g, five-week-old male mice). PBS, free Ce6 or nanophotosensitizers were intravenously (i.v.) injected via tail vein of mice when tumor diameter became larger than 4 mm. PBS was i.v. injected for control. For free Ce6, Ce6 dissolved in ethanol/Cremophor EL^®^ mixtures (1/1, *v*/*v*) was diluted with PBS (pH 7.4, 0.01 M) more than ten times. Nanophotosensitizers (ChitoPEITC-2 nanophotosensitizers, 40/4 in Table 2) were reconstituted in deionized water and then sterilized with 1.2 μm syringe filter. For empty nanoparticles, similar weight of empty ChitoPEITC-2 nanoparticles (vs. ChitoPEITC-2 nanophotosensitizers) was reconstituted in deionized water and then i.v. injected into the tail vein of the mice. Injection volume was 200 µL. For each group, five mice were used. Two days later, mice were anesthetized and then irradiated at 664 nm using the expanded homogenous beam (SH Systems, Gwangju, Korea) with 5.0 J/cm^2^. The day of first irradiation was determined as day 0, and 3 days later, mice were irradiated once more. The changes in tumor volume were measured with vernier calipers at 5 day intervals and were calculated with the following equation: tumor volume (mm^3^) = length × width^2^/2.

### 4.11. Statistical Analysis

The significance of statistical analysis was performed with Student’s *t* test using the SigmaPlot^®^ program (v. 11.0, Systat Software, Inc., Palo Alto, CA, USA). In statistical analysis, the minimum value of significance was estimated as *p* < 0.01.

## 5. Conclusions

ChitoPEITC conjugates were synthesized by conjugation of PEITC with the amine group of COS. ChitoPEITC nanophotosensitizers were fabricated by dialysis method. At TEM observation, ChitoPEITC nanophotosensitizers have spherical shapes with small hydrodynamic diameters of less than 200 nm. The higher PEITC contents in the ChitoPEITC copolymer resulted in a slower release rate of Ce6 from the nanophotosensitizers. Furthermore, the higher Ce6 contents resulted in a slower release rate of Ce6. In an intrinsic cytotoxicity study as dark toxicity, ChitoPEITC nanophotosensitizers have low toxicity against CCD986Sk cells and HCT-116 cells in the absence of light irradiation. In a cell culture study using HCT-116 cells, ChitoPEITC nanophotosensitizers showed a significantly higher Ce6 uptake ratio than that of free Ce6. Furthermore, cellular reactive oxygen species (ROS) production of nanophotosensitizers was significantly higher than that of free Ce6. PEITC and/or ChitoPEITC themselves contributed to the production of cellular ROS regardless of light irradiation. ChitoPEITC nanophotosensitizers showed significantly higher PDT efficacy against HCT-116 cells than that of free Ce6. These results indicate that ChitoPEITC nanophotosensitizers have superior potential in Ce6 uptake, ROS production and PDT efficacy. In the HCT-116-cell-bearing mice tumor-xenograft model, ChitoPEITC nanophotosensitizers delayed the changes in tumor volume rather than free Ce6. During the animal imaging study, ChitoPEITC nanophotosensitizers were concentrated in the tumor tissue, i.e., fluorescence intensity in the tumor tissue was stronger than that of other tissues. We suggest that ChitoPEITC nanophotosensitizers are a promising candidate for the treatment of human colon cancer cells.

## Figures and Tables

**Figure 1 ijms-23-13802-f001:**
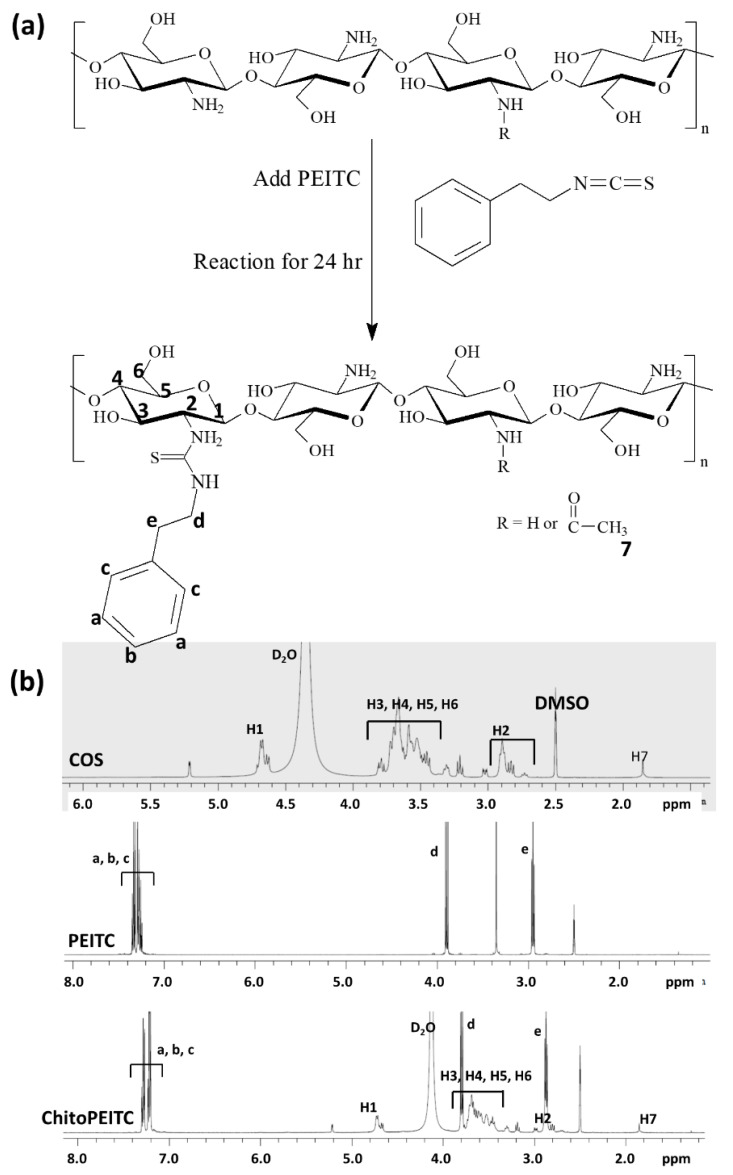
Synthesis scheme (**a**) and ^1^H NMR spectra (**b**) of COS, PEITC and ChitoPEITC conjugates. ChitoPEITC conjugation process was carried out at 20 °C for 24 h.

**Figure 2 ijms-23-13802-f002:**
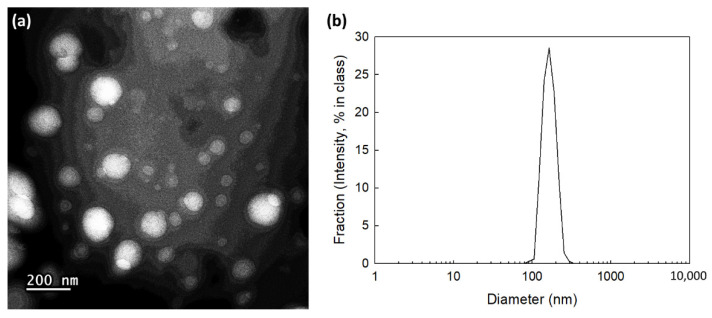
Morphological observations using TEM (**a**) and distribution of hydrodynamic diameter (**b**) of empty ChitoPEITC nanophotosensitizers (C hitoPEITC-2, 20:4 in Table 2).

**Figure 3 ijms-23-13802-f003:**
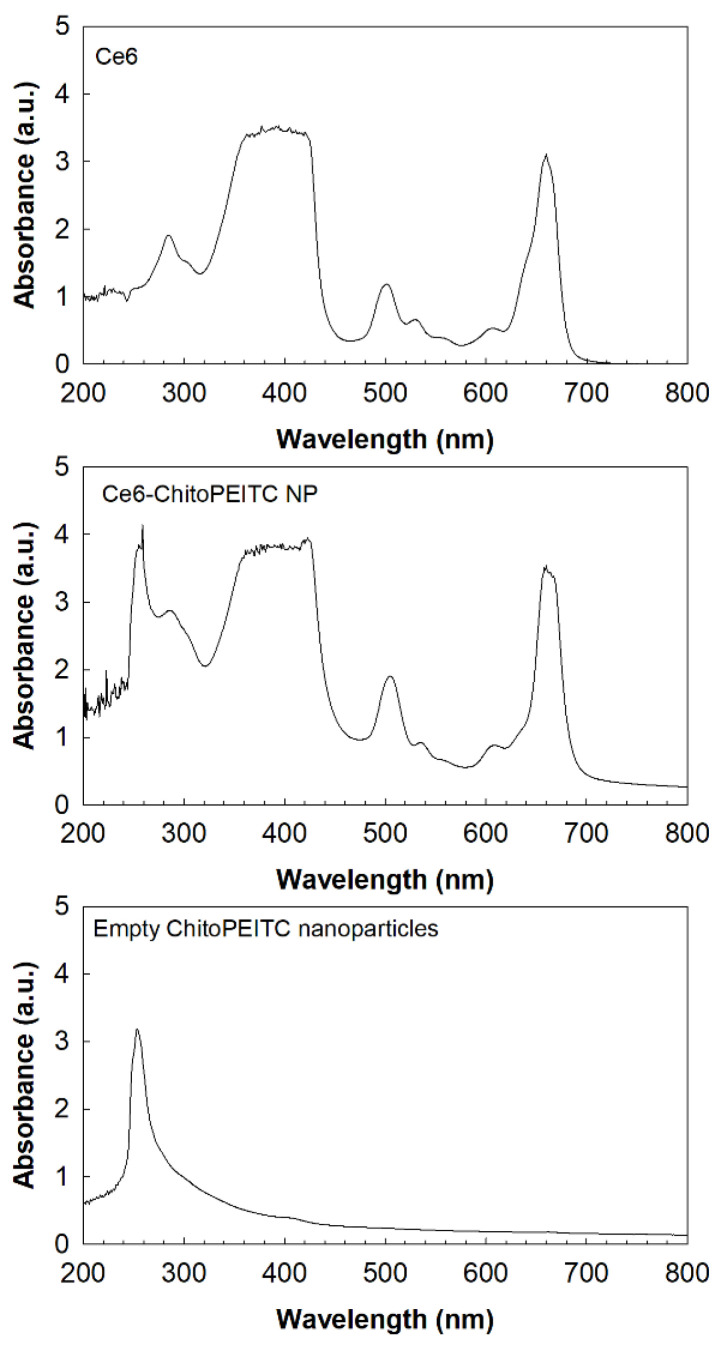
UV absorption spectra of Ce6, Ce6-incorporated CHitoPEITC nanophotosensitizers (Ce6-ChitoPEITC NP) and empty ChitoPEITC-2 nanoparticles (Empty CHitoPEITC NP). All chemicals were dissolved in DMSO:deionized water (1:1 mixtures). Ce6 concentration was adjusted to 0.1 mg/mL in Ce6 and Ce6-ChitoPEITC NP. For empty ChitoPEITC NP, empty ChitoPEITC-2 nanoparticles (1.0 mg/mL) were dissolved in DMSO:deionized water (1:1 mixtures).

**Figure 4 ijms-23-13802-f004:**
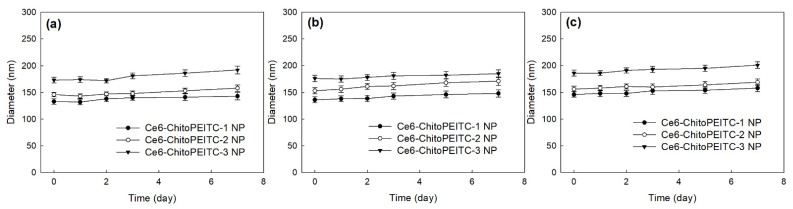
The stability of the hydrodynamic diameter of ChitoPEITC NP in various aqueous solutions. (**a**) Deionized water; (**b**) PBS (pH 7.4, 0.01 M); (**c**) RPMI-1640 media (10% fetal bovine serum, 1% antibiotics). Aqueous ChitoPEITC NP solution (1 mg/mL in deionized water in Table 2) in deionized water was diluted ten times with deionized water, PBS and RPMI-1640 media. All solutions were stored at 4 °C until measured. These results are average ± standard deviation (SD) from three different experiments.

**Figure 5 ijms-23-13802-f005:**
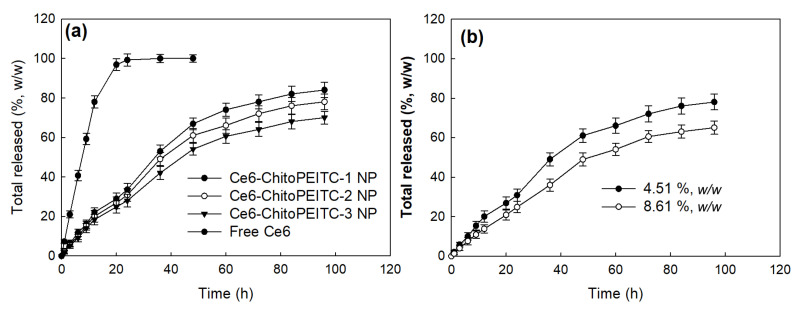
The effect of polymer (**a**) and Ce6 contents (**b**) on the release rate from nanophotosensitizers. These results are average ± standard deviation (SD) from three different experiments.

**Figure 6 ijms-23-13802-f006:**
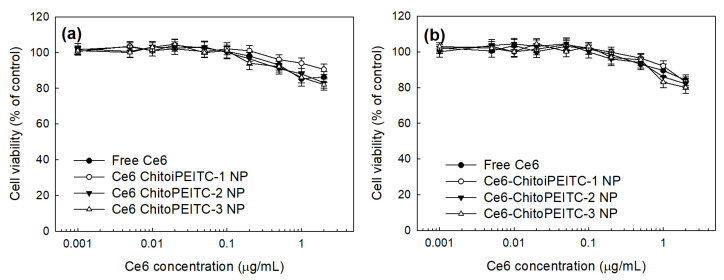
Intrinsic cytotoxicity as a dark toxicity of free Ce6 and nanophotosensitizers against CCD986Sk cells (**a**) and HCT-116 (**b**). These results are average ± SD from eight wells of 96-well plates.

**Figure 7 ijms-23-13802-f007:**
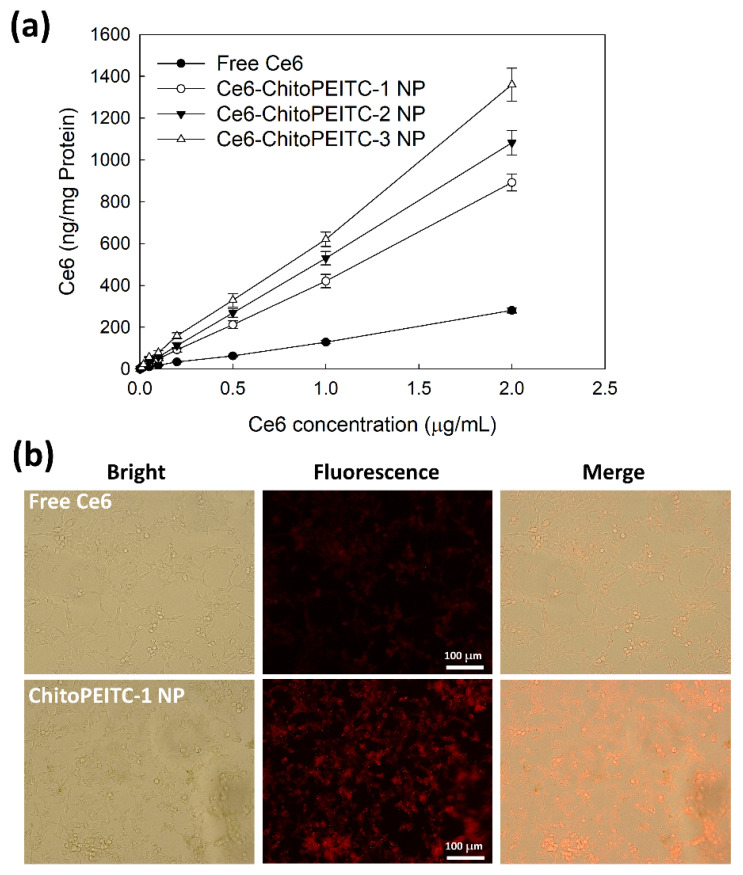
Ce6 uptake by HCT-116 cells (**a**) and fluorescence images (**b**) of HCT-116 cells. For fluorescence images, free Ce6 or Ce5-incorporated ChitoPEITC nanophotosensitizers (ChitoPEITC-1 NP, ChitoPEITC-2 NP, ChitoPEITC-3 NP) were treated to cells for 1 h and Ce6 concentration at 2 μg/mL. Magnification: 200×. Ce6 uptake results (Figure 6a shows average ± SD from eight wells of 96-well plates.

**Figure 8 ijms-23-13802-f008:**
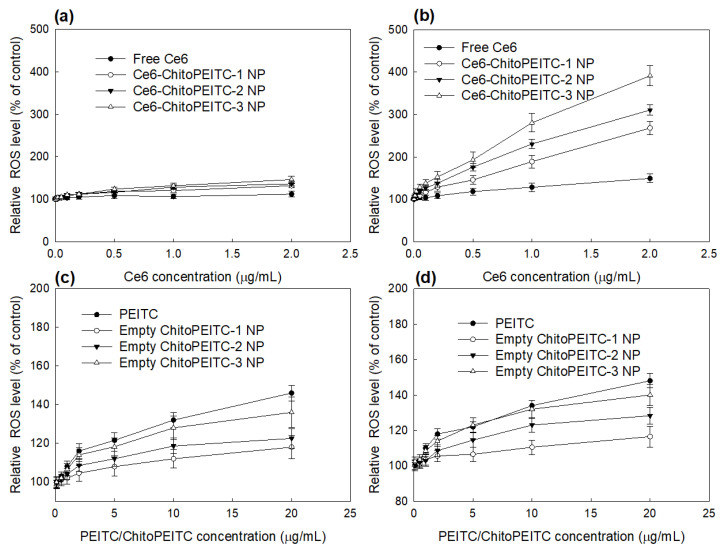
Cellular ROS generation of HCT-116 cells. Free Ce6 or Ce6-incorporated nanophotosensitizers were treated to cells, and then, cells were irradiated at 664 nm. (**a**) 0 J/cm^2^; (**b**) 2 J/cm^2^. PEITC or empty ChitoPEITC nanoparticles were treated to cells and then irradiated at 664 nm. (**c**) 0 J/cm^2^; (**d**) 2 J/cm^2^. ChitoPEITC NP = ChitoPEITC nanophotosensitizers. These results are average ± SD from eight wells of 96-well plates.

**Figure 9 ijms-23-13802-f009:**
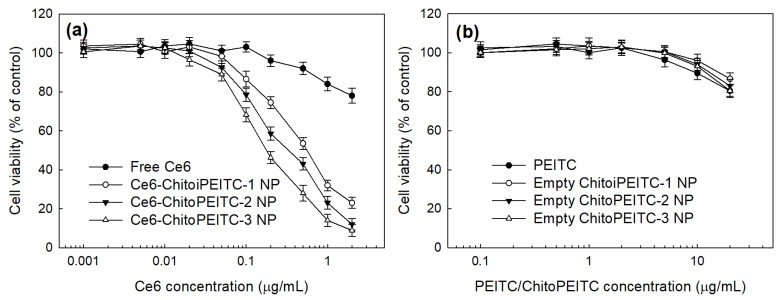
PDT efficacy of free Ce6 and nanophotosensitizers. (**a**) Free Ce6 or Ce6-incorporated nanophotosensitizers; (**b**) PEITC and empty nanoparticles as ChitoPEITC copolymer. Cells were irradiated at 2 J/cm^2^. ChitoPEITC NP = ChitoPEITC nanophotosensitizers. These results are average ± SD from eight wells of 96-well plates.

**Figure 10 ijms-23-13802-f010:**
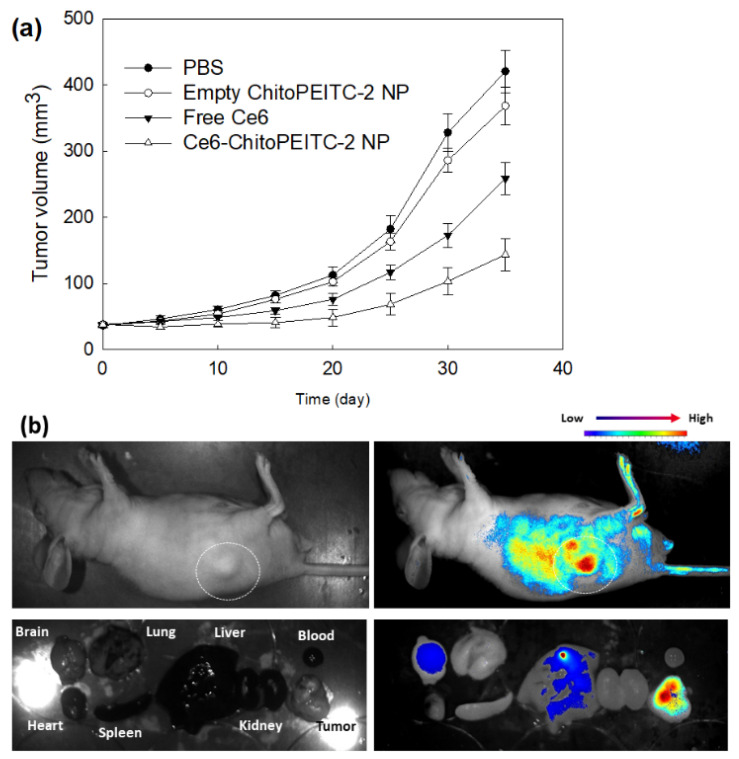
PDT efficacy against HCT-116-bearing mice and animal tumor imaging. (**a**) The effect of PDT on HCT-116 tumor growth. (**b**) Animal tumor imaging using HCT-116 tumor xenograft. Free Ce6 or nanophotosensitizer (ChitoPEITC-2, 40/4 in Table 2) solution (10 mg Ce6/kg) was i.v. administered via the tail vein of the mice. PBS and empty ChitoPEITC-2 nanoparticles were injected for control and empty nanoparticle treatment. For fluorescence imaging, nanophotosensitizers were i.v. injected via the tail vein of mice. One day later, mice were anesthetized and then irradiated at 664 nm (5.0 J/cm^2^). For each group, five mice were used, which are expressed as average ± SD. Empty ChitoPEITC-2 NP vs. Ce6-ChitoPEITC-2 NP, *p* < 0.01; Free Ce6 vs. Ce6-ChitoPEITC-2 NP, *p* < 0.01.

**Table 1 ijms-23-13802-t001:** Characterization of CHitoPEITC conjugates.

	Feeding Ratio (Glucosamine: PEITC, mM/mM)	DS of PEITC ^a^	Hydrodynamic Diameter (nm) ^b^	PDI ^c^	Zeta Potential (mV) ^d^
ChitoPEITC-1	2.0:0.2	8.6	89 ± 4	0.096	13 ± 0.5
ChitoPEITC-2	2.0:0.4	18	113 ± 5	0.116	9.6 ± 0.4
ChitoPEITC-3	2.0:0.8	36	126 ± 4	0.109	7.1 ± 0.3

^a^ Substitution degree (DS) of PEITC (PEITC number/100 glucosamine) was estimated from ^1^H NMR spectra in Figure 1. DS of PEITC was calculated as follows: [(Integral value of e proton/2)/(Integral values of C1 proton of COS)] × 100. ^b^ Hydrodynamic diameters are average ± standard deviation from three different measurement. ^c^ Polydispersity index (PDI). ^d^ Zeta potential is average ± standard deviation (SD) from three independent measurements.

**Table 2 ijms-23-13802-t002:** Characterization of ChitoPEITC nanophotosensitizers.

ChitoPEITC/Ce6 Weight Ratio (mg/mg)	Drug Contents (%, *w*/*w*) ^a^	Loading Efficiency (%, *w*/*w*) ^a^	Hydrodynamic Diameter (nm) ^b^	PDI	Zeta Potential (mV) ^c^
Ce6-ChitooPEITC-1 NP ^d^					
40/2	4.4	93	133 ± 5	0.114	5.9 ± 0.4
Ce6-ChitoPEITC-2 NP					
40/2	4.5	95	146 ± 4	0.108	4.2 ± 0.3
40/4	8.6	95	167 ± 6	0.121	2.8 ± 0.4
Ce6-ChitoPEITC-3 NP					
40/2	4.6	97	173 ± 5	0.113	2.7 ± 0.4

^a^ Drug content (*w*/*w*, %) = (Ce6 weight/total weight of nanophotosensitizers) × 100. Loading efficiency (*w*/*w*, %) = (Ce6 weight in the nanophotosensitizers/feeding weight of Ce6) × 100. ^b^ Particle sizes are average ± standard deviation from three different measurements. ^c^ Zeta potential is average ± standard deviation (SD) from three independent measurements. ^d^ NP means nanophotosensitizers.

## Data Availability

Not applicable.

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
