# Peer review of "Phenethyl Isothiocyanate-Conjugated Chitosan Oligosaccharide Nanophotosensitizers for Photodynamic Treatment of Human Cancer Cells"

_ijms, 2022, doi:10.3390/ijms232213802_

Round 1

Reviewer 1 Report

The paper describe a method of production and use of a novel nanoparticles loaded with a photosensitizer Ce6 named Chito-PEITC-nanophotosensitizer, for colorectal cancer treatment using photodynamic therapy. The authors used different very complementary approaches with several experiments ranging from in vitro to in vivo. The results obtained seem promising.  This manuscript is of good quality but can be improved :

The photophysical characterization of the different compounds was not reported, especially absorption and emission spectrum, fluorescence yield and half-life. The authors used fluorescence intensity to compare the uptake of each compound, so it should be good to check that the photophysical parameter of Ce6 were not modified after absorption onto ChitoPEITC conjugate.

The authors have evaluated the dark toxicity of compounds with normal cells (human skin fibroblast cells) and tumor (Human colon cancer cells) and only photo toxicity with cancer cells. It will be interesting to have these values.

The authors have evaluated the “intracellular Ce6 uptake” of compounds by cells (line 161) but they didn’t eliminate membrane of cells. Furthermore the authors used fluorescence microscopy (figure 5b) but not confocal fluorescence microscopy, then the authors cannot conclude that these compounds have been incorporated into cells. It would be better to eliminate the word “intracellular” from the manuscript to avoid misinterpretation.

For some experimentations of the same type, the authors used a different x axis onto graph : Ce6 concentration and PEITC/Chito/PEITC concentration (Figure 6 and 7). I did not find any information in the manuscript to explain the corresponding factor between Ce6 concentration and PEITC/ChitoPEITC concentration ; is-it really a factor 10? This information must be added in the material and method section.

The result obtained using fluorescence imaging for animal isn’t clear (figure 10b): is the image of HCT-116 treated with free Ce6 is in the left? and ChitoPEITC-2 in the right?; top view, side view… Why is the tumor not located in the same place on the top and bottom image ?  Why the authors used ChitoPEITC-2 while ChitoPEITC-3 formulation seemed more promising.

The quality of images needs to be improved :

-          Figure 2a: is-it really a TEM observation? Usually objects are visible in black on a white background. Here the objects are blurred, we don’t see well the morphology of the nanoparticles.

-          Figure 5b: cells weren’t visible, it looks like there are aggregates on the surface of the cells. The authors must add the images of control cells (without Ce6 or nanophotosensitizer) to show the intrinsic autofluorescence level of cells.

-          Scale barre is missing on Figure 2a and 5b.

The authors didn’t explain why they use PEITC in the introduction part, while a paragraph is devoted to the role of ROS in PDT as well as their regulatory mechanism… ; the reader have to wait to the discussion to find out. Same remark for chitosan. Furthermore a study using ChitoCe6 nanophotosensitizer has been published in 2017 (reference 41) and wasn’t raise in the introduction part. The authors could also compare their results with those obtained in this study in the discussion section.

Finally, the authors didn’t indicate the number of independent experiment used in each study.

Materials and methods:

4.5 The authors have evaluated the size of the NP using Nano-zetasizer. This device is able to measure the hydrodynamic diameter of NPs not the particle size. The author can use TEM to calculate the particle size of the NPs. To be clarified in the material and method section.

4.7 did the authors have add glutamine into culture media ? this supplement is generally used to the culture of mammalian cells.

4.8 why the authors have filtered the aqueous solution of their nanophotosensitizer?

Line 373-374: why the authors dilute their NPs in serum-free media while they injected the product intravenously into the animal?

Line 375-379 : It is noted that after washing, serum-free media was added onto cells and incubated for 24h in a CO2 after PDT (incubator is missing)… not culture media?

4.9 h=hour

The authors didn’t eliminate membrane. Then they cannot evaluate the intracellular Ce6 uptake , then measure the incorporation of compounds into cell + compounds adsorbed to the membrane of cells. To be clarified.

Did the authors have verified the photophysical properties of their compounds / Ce6? If not, they cannot used fluorescence intensity to compare the incorporation of nanophotosensitizer with Ce6 alone, they should realize for each compound a calibration curve using known concentration.

No information about fluorescence observation parameters (time exposure, power or gain, wavelength…).

4.10: in vivo : use italic

Line 429 : solution cannot be sterilized using 1.2 µm syringe filter.

The number of animal used in different groups weren’t indicated.

Conclusion:

Line 458: Chito-PEITC nanophotosensitizer didn’t inhibit the growth of tumor. As seen in figure 10a, the growth of cells treated with nanophotosensitizer + PDT is inhibit before 20 days, but tumor volume increase after this time point.

Author Response

Response to Reviewer 1’s comment

The paper describe a method of production and use of a novel nanoparticles loaded with a photosensitizer Ce6 named Chito-PEITC-nanophotosensitizer, for colorectal cancer treatment using photodynamic therapy. The authors used different very complementary approaches with several experiments ranging from in vitro to in vivo. The results obtained seem promising.  This manuscript is of good quality but can be improved :

Answer) Thanks for your comment. According to your comments, werevised manuscript fullty.

The photophysical characterization of the different compounds was not reported, especially absorption and emission spectrum, fluorescence yield and half-life. The authors used fluorescence intensity to compare the uptake of each compound, so it should be good to check that the photophysical parameter of Ce6 were not modified after absorption onto ChitoPEITC conjugate.

Answer) Thanks for your comment. At this moment, we focused onto additive effect of PEITC and ChitoPEITC for ROS formation Anyway, we added UV-VIS spectra of Ce6 and ChitoPEITC NP. Practically, we measured fluorescence spectra many times in a previous report. The problem is that fluorescence emission spectra showed completely similar patterns between 500 nm ~800 nm (excitation wavelength: 400nm) (even though its intensity is different) regardless of polymer or carrier materials (Int. J. Mol. Sci. 2022, 23, 3117; Cells 2021, 10, 2190). Practically, we afraid fluorescence emission spectra looks like similar data even though we measured separately. Then, we added UV-spectra rather than fluorescence emission data. Please consider our effort for this study.

 Thanks.

Results

To analyze ultraviolet-visible (UV) spectra of nanophotosensitizers, free Ce6, Ce6-incorporated nanophotosensitizers and empty nanoparticles were measured with UV-VIS spectrophotometer as shown in Figure 3. As shown in Figure 3, free Ce6 showed specific peak intensity between 200 nm ~ 800 nm and maximum peaks were observed at 664 nm. Ce6-incorporated nanophotosensitizers almost similar peak intensity between 300 nm ~ 800 nm while no specific peak intensity was observed in the field with empty nanoparticles. These results indicated that no chemical modification of Ce6 itself occurred during fabrication process of nanophotosensitizers.

Figure 3. UV absorption spectra of Ce6, Ce6-incorporated CHitoPEITC nanophotosensitizers (Ce6-ChitoPEITC NP) and empty ChitoPEITC-2 nanoparticles (Empty CHitoPEITC NP). All chemicals were dissolved in DMSO:deionized water (1:1 mixtures). Ce6 concentration was adjusted to 0.1 mg/mL in Ce6 and Ce6-ChitoPEITC NP. For empty ChitoPEITC NP, empty ChitoPEITC-2 nanoparticles (1.0 mg/mL) were dissolved in DMSO:deionized water (1:1 mixtures).   

The authors have evaluated the dark toxicity of compounds with normal cells (human skin fibroblast cells) and tumor (Human colon cancer cells) and only photo toxicity with cancer cells. It will be interesting to have these values.

Answer) Thanks for your comment. Practically, both of Ce6 and ChitoPEITC must have photo-toxicity against normal cells as well as cancer cells. That is the reason for laser irradiation only in the tumor site. If laser irradiate normal tissues, normal tissues must have burning effect. As you know, patients, which is administered with normal photosensitizers, should be protected from light for a while.

The authors have evaluated the “intracellular Ce6 uptake” of compounds by cells (line 161) but they didn’t eliminate membrane of cells. Furthermore the authors used fluorescence microscopy (figure 5b) but not confocal fluorescence microscopy, then the authors cannot conclude that these compounds have been incorporated into cells. It would be better to eliminate the word “intracellular” from the manuscript to avoid misinterpretation.

Answer) Thanks for your comment. According to your comment, we used cellular uptake rather than intracellular uptake in the manuscript. Furthermore, we provided better images in Figure 6. Please consider our effort. Thanks.

For some experimentations of the same type, the authors used a different x axis onto graph : Ce6 concentration and PEITC/Chito/PEITC concentration (Figure 6 and 7). I did not find any information in the manuscript to explain the corresponding factor between Ce6 concentration and PEITC/ChitoPEITC concentration ; is-it really a factor 10? This information must be added in the material and method section.

Answer) Thanks for your comment. Typically, in our manuscript, ChitoPEITC-1 NP means Ce6-incroporated ChitoPEITC nanoparticles and ChitoPEITC-1 means empty nanoparticles of ChitoPEITC-1. To avoid these confusion, we changed the mansucript as Ce6-incorportaed NP or empty NP.

The result obtained using fluorescence imaging for animal isn’t clear (figure 10b): is the image of HCT-116 treated with free Ce6 is in the left? and ChitoPEITC-2 in the right?; top view, side view… Why is the tumor not located in the same place on the top and bottom image ?  Why the authors used ChitoPEITC-2 while ChitoPEITC-3 formulation seemed more promising.

Answer) Thanks for your comment. Practically, the reason that we used ChitoPEITC nanoparticles because of ChitoPEITC-3 nanoparticles are bigger than ChitoPEITC-2 while Chitopeitc-1 nanoparticles have low drug contents. Then we wanted nice targeting effect with small size with reasonable contents of Ce6. Then we choose ChitoPEITC-2. Thanks for your comments.

The quality of images needs to be improved :

Answer) Thanks for your comment. According to your comment, we provided and revised with better TEM image and cell images as shown in Figure 2 and 6.

-          Figure 2a: is-it really a TEM observation? Usually objects are visible in black on a white background. Here the objects are blurred, we don’t see well the morphology of the nanoparticles.

Answer) Thanks for your comment. As indicated in the Materials and methods section (4.5 Characterization of nanophotosensitizers), we used Phosphotungstic acid (0.1% w/w in deionized water) for negative staining of nanophotosensitizers. When phosphotungstic acid was used, nanoparticles showed as a white images and the background showed as black. Thanks for your comment.

-          Figure 5b: cells weren’t visible, it looks like there are aggregates on the surface of the cells. The authors must add the images of control cells (without Ce6 or nanophotosensitizer) to show the intrinsic autofluorescence level of cells.

Answer) Thanks for your comment. According to your comment, we observed again with fluorescence microscopy and indicated better images.

-          Scale barre is missing on Figure 2a and 5b.

Answer) Thanks for your comment. According to your comment, we added scale bar. Thanks for your comment.

The authors didn’t explain why they use PEITC in the introduction part, while a paragraph is devoted to the role of ROS in PDT as well as their regulatory mechanism… ; the reader have to wait to the discussion to find out. Same remark for chitosan. Furthermore a study using ChitoCe6 nanophotosensitizer has been published in 2017 (reference 41) and wasn’t raise in the introduction part. The authors could also compare their results with those obtained in this study in the discussion section.

Answer) Thanks for your comment. According to your comment, we fully revised the manuscript and added reference previous research and added introduction of PEITC.

Introduction section

Especially, Jia et al., reported that phenethyl isothiocyanate (PEITC) can be used as a ROS-inducing agent since PEITC inhibits the ROS scavenger and then sinergistically enhances the cytotoxic effects of BMN 673 through overproduction of ROS and then induces DNA damage and apoptosis [20]. Liu et al., also reported that PEITC efficiently depletes L-glutathione in the cells, which is ROS scavenger, and then over-production of ROS following with synergistic death of cancer cells [21]. 

  1. Jia, Y.; Wang, M.; Sang, X.; Liu, P.; Gao, J.; Jiang, K.; Cheng, H. Phenethyl isothiocyanate enhances the cytotoxic effects of PARP inhibitors in high-grade serous ovarian cancer cells. Oncol. 2022, 11, 812264.
  2. Liu, Q.; Ding, X.; Xu, X.; Lai, H.; Zeng, Z.; Shan, T.; Zhang, T.; Chen, M.; Huang, Y.; Huang, Z.; Dai, X.; Xia, M.; Cui, S. Tumor-targeted hyaluronic acid-based oxidative stress nanoamplifier with ROS generation and GSH depletion for antitumor therapy. J. Biol. Macromol. 2022, 207, 771-783. 

Discussion section

Additionally, nanoparticles are regarded to be a suitable carrier for absorption of photosensitizers against epithelial cancer or squamous cancer cells while free photosensitizers revealed low uptake ratio against tumor and immediately removed from the target site [47]. In this study, Jeong et al., reported that Ce6-chitosan complexes also improve Ce5 uptake/ROS generation/PDT efficacy against gastrointestinal (GI) cells and, furthermore, they efficiently target tumor tissues while  free Ce6 rapidly disappeared.

  1. Jeong, Y.I.; Cha, B.; Lee, H.L.; Song, Y.H.; Jung, Y.H.; Kwak, T.W.; Choi, C.; Jeong, G.W.; Nah, J.W.; Kang, D.H. Simple nanophotosensitizer fabrication using water-soluble chitosan for photodynamic therapy in gastrointestinal cancer cells. J. Pharm. 2017, 532, 194-203.

Finally, the authors didn’t indicate the number of independent experiment used in each study.

 Answer) Thanks for your comment. According to your comment, we indicated it in the Figure caption. Thanks.

Materials and methods:

4.5 The authors have evaluated the size of the NP using Nano-zetasizer. This device is able to measure the hydrodynamic diameter of NPs not the particle size. The author can use TEM to calculate the particle size of the NPs. To be clarified in the material and method section.

Answer) Thanks for your comment. According to your comment, we added the number of study. At this moment, we used TEM for confirmation morphology of nanoparticles rather than analyzed particle sizes. We indicated hydrodynamic radius in the Materials and methods section.

4.7 did the authors have add glutamine into culture media ? this supplement is generally used to the culture of mammalian cells.

Answer) Thanks for your comment. At this moment, we are going to do additives effects or synergistic effects with various ROS scavenger such as L-glutathione, N-acetyl cysteine and natural antioxidants. Furthermore, we will also investigate synergistic effect of natural ROS-producing agents (Such as piperlongumine) and GSH-scavenger (PEITC). And glutamate is also going to investigate to the mechanism of GSH scavenging mechanism. Please consider this situation. We would like to publish these results as in other article

4.8 why the authors have filtered the aqueous solution of their nanophotosensitizer?

Answer) Thanks for your comment. We filtered nanophotosensitizers with for cell culture and 1.2 um for cell culture and animal study. This size filter only completely filtered without loss of nanoparticles, o changes of drug contents, no cell contamination and negligible particle sizes. We indicated it in the Materials and methods section.

Line 373-374: why the authors dilute their NPs in serum-free media while they injected the product intravenously into the animal?

Answer) Thanks for your comment. Practically, there is no special reason for use of serum-free media. Normally, we used PBS for IV injection into mouse. In our experience for animal study, nanoparticles were stable during movement/wobble of nanophotosensitizer solution from lab to animal experiment lab because animal experiment lab is far from our lab (about 30 min). In the case of serum, we afraid that serum may have immunological problem against mouse and then we used serum-free media. Another reason is that we used serum-free media for IV injection of nanophotosensitizers because we used media for in vitro cell culture experiment. We want same situation against cancer cells in vitro and in vivo.

Line 375-379 : It is noted that after washing, serum-free media was added onto cells and incubated for 24h in a CO2 after PDT (incubator is missing)… not culture media?

Answer) Thanks for your comment. It is right. We used 2 × 10^4 cells/well in 96 well with serum-free media to avoid over-growth of cells. AS far as our experience, when 3 × 10^4 cells/well in 96 well are seeded, cells were closely packed in the well. If we use culture media, control cells (without treatment of Ce6) must be over-packed closely and treatment protocol is hard to trustable. Another reason for serum-free media is that we want cytotoxicity by PDT instead of growth inhibition by PDT. Please consider our efforts. Thanks.

4.9 h=hour

Answer) Thanks for your comment. According to your comment, we corrected it.

The authors didn’t eliminate membrane. Then they cannot evaluate the intracellular Ce6 uptake , then measure the incorporation of compounds into cell + compounds adsorbed to the membrane of cells. To be clarified.

Answer) Thanks for your comment. According to your comment, we observed again of fluorescence images of cell after treatment of Ce6 or nanophotosensitizers. Practically, we washed with PBS two times before measurement of Ce6 uptake ratio. We indicated it in the experimental section. Thanks again for your comment.

Did the authors have verified the photophysical properties of their compounds / Ce6? If not, they cannot used fluorescence intensity to compare the incorporation of nanophotosensitizer with Ce6 alone, they should realize for each compound a calibration curve using known concentration.

Answer) Thanks for your comment. We already reported the calibration curve using UV-VIS spectrophotometer in recent publication. We reported the calibration curve of Ce6 in the report of (Int. J. Mol. Sci. 2022, 23, 3594.). We measured always checked calibration curve. If we added calibration curve of Ce6 in various media, we afraid that those data must be problem because we already reported. Anyway, we indicated how to measure Ce6 contents in the materials and methods section with reference. Please consider our situation. Thanks.

In the experimental section

To calculate Ce6 concentration, calibration curve was determined as follows [48]: 10 mg of Ce6 was dissolved in 10 ml of DMSO (1 mg/ml). This solution was diluted with deionized water/DMSO mixtures (1/9, v/v) 100 times. Genesys 10s UV-VIS spectrophotometer (Thermo Fisher Scientific, Waltham, Massachusetts, USA) was employed to measure absorption spectra of this solution. Calibration curve was determined from 0.1 to 7 µg/ml of Ce6. Otherwise, Ce6 (1mg/ml DMSO) was 30 times diluted with PBS for estimation of Ce6 concentration in the PBS and then calibration curve was determined at the range of 0.3 ~ 10 µg/ml of Ce6 concentration.

References

Yoon, J.; Kim, H.; Jeong, Y.-I.; Yang, H.S. CD44 Receptor-Mediated/Reactive Oxygen Species-Sensitive Delivery of Nanophotosensitizers against Cervical Cancer Cells. Int. J. Mol. Sci. 202223, 3594.

No information about fluorescence observation parameters (time exposure, power or gain, wavelength…).

Answer) Thanks for your comment. Practically, we observed cell fluorescence images with fluorescence microscopy because we have no confocal microscope in our department. Of course, other university has confocal microscope but we should be move more than one hour and we afraid that fluorescence photophysical properties must be decreased or changed. Please consider our situation at this moment. We indicated fluorescence microscopic measurement (Eclipes 80i; Nikon, Tokyo, Japan) in the materials and methods more detailed. In this device, we found that red filter for observation of Ce6 was (Application: Alexa 594; Emission Bandwidth: 600-660 nm) in the specification sheet. Anyway we indicated it in the experimental section

Fluorescence observation: HCT-116 cells (2 × 105) seeded on the cover glass in six well plates were cultured overnight. These cells were washed with PBS and then treated with Ce6 or nanophotosenstitizers. One hour later, cells were washed with PBS twice and then fixed with 4 % paraformaldehyde (PFA) solution in PBS for 15 m. After that, cells were immobilized with immobilization solution (Immunomount, Thermo Electron Co. Pittsburgh, PA, USA). Observation of cells were carried out with a fluorescence microscope (Emission bandwidth: 600-660 nm) (Eclipes 80i; Nikon, Tokyo, Japan). For observation of Ce6, red filter was used and bright field also observed.

4.10: in vivo : use italic

Answer) Thanks for your comment. According to your comments. We revised the manuscript. According to your comment, we corrected in vivo as a italic.

Line 429 : solution cannot be sterilized using 1.2 µm syringe filter.

Answer) Thanks for your comment. Practically, we filtered nanophotosensitizers with 1.2 µm syringe filter. This size filter only completely filtered without loss of nanoparticles, no changes of drug contents, no cell contamination and negligible particle sizes. We indicated it in the Materials and methods section. AT least 7 days, we did not find contamination in vitro cell culture experiments.

The number of animal used in different groups weren’t indicated.

Answer) Thanks for your comment. As you commented, we indicated in the fugure caption. For each group, five mice were used and expressed as average ± S.D.

Conclusion:

Line 458: Chito-PEITC nanophotosensitizer didn’t inhibit the growth of tumor. As seen in figure 10a, the growth of cells treated with nanophotosensitizer + PDT is inhibit before 20 days, but tumor volume increase after this time point.

Answer) Thanks for your comment. I agree to your opinion. According to your comment, we revised the manuscript.

In results section

PDT efficacy of nanophotosensitizers were investigated with HCT-116 bearing mice as shown in Figure 10. As shown in Figure 10(a), tumor volume was gradually increased with time course. When free Ce6 or nanophotosensitizers were injected with light irradiation, the tumor volume was relatively smaller than those of control or empty nanoparticles at 30 days. Practically, empty nanoparticles did not affect to the changes of tumor volume. Among all treatments, nanophotosensitizers properly delayed the changes of tumor volume compared to free Ce6 and empty nanoparticles as shown in Figure 10(a). Figure 10(b) supports these results, i.e. strong fluorescence intensity was observed in tumor xenograft at whole body imaging of mouse. Furthermore, fluorescence images of organs showed that fluorescence intensity was strongest in the tumor tissues compared to other organs. These results indicated that ChitoPEITC nanophotosensitizers can be efficiently accumulated in the tumor tissue and then induce superior PDT efficacy.

In conclusion section

In HCT-116 cell bearing mice tumor-xenograft model, ChitoPEITC nanophotosensitizers delayed the changes of tumor volume rather than free Ce6.

Reviewer 2 Report

Specific comments:

1. The authors use the same acronym (ChitoPEITC) for conjugates, loaded and unloaded particles. This is very confusing when reading the paper. I would recommend that the authors use different acronyms for them (e.g. ChitoPEITC, ChitoPEITC NPs, ChitoPEITC-Ce6 NPs). For example, in Figures 3-10 it is not clear at all what samples are referred to, taking into account the different Ce6 loading.

2. Sections 2.1 and 2.2: It would be useful to add the zeta potential values of loaded and unloaded ChitoPEITC nanoparticles to characterize the stability of the dispersions.

3. Line 102: Add the formula for calculating the degree of substitution (DS) from the NMR spectra to the text.

4. Line 103: I would ask the authors to add a short reasoning on why the ChitoPEITC conjugates form nanoparticles.

5. Figure 1a: Draw the structures of chitosan and its derivative, avoiding the use of angles around the glycosidic bond (the angle in the structural formula denotes a carbon atom). Add the notation a-e protons to the substituent. Also add the reaction temperature in addition to the reaction time.

6. Figures 1a and 1b are hardly visible. I would recommend increasing their size and resolution.

7. Table 1: Round off the DS values to two significant figures.

8. Table 1: Replace "particle size" with "hydrodynamic diameter" in Table 1 and in the text if these values were determined by dynamic light scattering on the Nano-Zetasizer instrument. Add polydispersity index (PDI) values here and discuss them in the text.

9. Tables 1-2 and throughout the text: The standard deviation should be expressed as ONE significant figure; that is, unless the number is between 11 and 19 times some power of ten, in which case you can use two significant figures. The mean value should be rounded off at the decimal place corresponding to the last significant digit of its standard deviation. E.g., 84.4±4.5 (Table 1) should be presented as 84±5, etc.

10. Section 2.2: To characterize drug-loaded nanoparticles, data on loading efficiency and encapsulation efficiency are usually provided. I ask you to add this data to the paper.

11. Line 135: I wouldn't call it "burst drug release" because there is almost linear release up to 50 hours.

12. Figure 3: Add the results of the blank experiment (dialysis of pure Ce6) to exclude any artifacts related to dissolution of Ce6 followed by diffusion of Ce6 through the dialysis membrane.

13. Line 295: The chitosan sample must be thoroughly characterized regarding its molecular weight (by viscometry, light scattering, or size exclusion chromatography) and the degree of deacetylation (by NMR, IR, elemental analysis, or titration). The properties of chitosan are very dependent on these parameters; therefore, the wide ranges of values provided by the manufacturer are clearly insufficient. Also indicate the source of chitosan (crab, shrimp, fungi, etc.).

14. Line 327: The authors performed dialysis of loaded nanoparticles for 1 day with frequent water changes. With this methodology, I would have assumed a significant loss of Ce6 drug (about 30% as well as during 24 hours of release, Figure 3), however, according to Table 2, more than 93% of the drug remained encapsulated. How would the authors explain this phenomenon?

15. Lines 330-335: It is not clear from this methodology how the complete 100% release of Ce6 from the nanoparticles was achieved? Please clarify.

Author Response

Response to Reviwer 2’s comment

Specific comments:

  1. The authors use the same acronym (ChitoPEITC) for conjugates, loaded and unloaded particles. This is very confusing when reading the paper. I would recommend that the authors use different acronyms for them (e.g. ChitoPEITC, ChitoPEITC NPs, ChitoPEITC-Ce6 NPs). For example, in Figures 3-10 it is not clear at all what samples are referred to, taking into account the different Ce6 loading.

Answer) Thanks for your comment. According to your comment, we corrected its expression. For example, CHitoPEITC nanoparticles à Empty ChitoPEITC NP or Empty ChitoPEITC nanoparticles; Ce6-incorporated ChitoPEITC NP à Ce6-ChitoPEITC NP. Thanks.

  1. Sections 2.1 and 2.2: It would be useful to add the zeta potential values of loaded and unloaded ChitoPEITC nanoparticles to characterize the stability of the dispersions.

Answer) Thanks for your comment. According to your comment, we measured zeta potential three different times. Furthermore, to analyze stability of nanoparticles, changes of particle size was added in

Table 1. Tables should be placed in the main text near to the first time they are cited.

Feeding ratio (glucosamine:PEITC, mM/mM)

DS of PEITC a

Hydrodynamic radius (nm) b

PDI c

Zeta potential (mV) d

ChitoPEITC-1

ChitoPEITC-2

ChitoPEITC-3

2.0:0.2

2.0:0.4

2.0:0.8

8.6

17.8

35.6

89.4±4.3

112.9±5.2

126.2±4.1

0.096

0.116

0.109

12.8±0.53

9.6±0.41

7.1±0.32

a Substitution degree (DS) of PEITC (PEITC number/100 glucosamine) was estimated from 1H NMR spectra in Figure 1.

b Particles sizes were average ± standard deviation from three different measurement.

c Polydispersity index (PDI).

d Zeta potential was average ± standar deviation (S.D.) from three independent measurement.

Table 2. Characterization of ChitoPEITC nanophotosensitizers.

ChitoPEITC/Ce6 weight ratio (mg/mg)

Drug contents (%, w/w) a

Particle size (nm) b

PDI

Zeta Potential (mV) c

Theoretical

Experimental

Ce6-ChitooPEITC-1 NP c

40/2

Ce6-ChitoPEITC-2 NP

40/2

40/4

Ce6-ChitoPEITC-3 NP

40/2

4.76

4.76

9.09

4.76

4.43

4.51

8.61

4.64

133±5.2

146±4.1

167±5.8

173±5.3

0.114

0.108

0.121

0.113

5.9±0.43

4.2±0.31

2.8±0.43

2.7±0.36

a Drug content (w/w, %) = (Ce6 weight/total weight of nanophotosensitizers) × 100.

b Particles sizes were average ± standard deviation from three different measurement.

c Zeta potential was average ± standar deviation (S.D.) from three independent measurement.

d NP means nanophotosensitizers.

Figure 4. The stability of hydrodynamic radius of ChitoPEITC NP in deionized water (a), PBS (pH 7.4, 0.01 M) and RPMI1-1640 media (10 % fetal bovine serum, 1 % antibiotics) (c). Aqueous ChitoPEITC NP solution (1 mg/mL in deionized water in Table 2) in deionized water was diluted ten times with deionized water, PBS and RPMI-1640 media. All solutions were stored in the 4 oC until measured them. These results were average ± standard deviation (SD) from three different experiment.

  1. Line 102: Add the formula for calculating the degree of substitution (DS) from the NMR spectra to the text.

Answer) Thanks for your comment. According to your comment, we added the formula of DS in the Table 1. Practically, we measured NMR spectra several times and used to calculate DS of PEITC.

a Substitution degree (DS) of PEITC (PEITC number/100 glucosamine) was estimated from 1H NMR spectra in Figure 1. DS of PEITC was calculated as follows: [(Integral value of e proton/2)/(Integral values of C1 proton of chitosan)] × 100.

  1. Line 103: I would ask the authors to add a short reasoning on why the ChitoPEITC conjugates form nanoparticles.

Answer) Thanks for your comment. According to your comment, we added short comment about nanoparticle formation of ChitoPEITC.

ChitoPEITC conjugates are able to form nanoparticles because PEITC has a hydrophobic properties and chitosan is a water-soluble molecule. These properties must be induced aggregates as a nano-sized vehicle in the aqueous solution. Therefore, particle size was measured whether or not ChitoPEITC conjugates can form nanoparticles as shown in Table 1.

  1. Figure 1a: Draw the structures of chitosan and its derivative, avoiding the use of angles around the glycosidic bond (the angle in the structural formula denotes a carbon atom). Add the notation a-e protons to the substituent. Also add the reaction temperature in addition to the reaction time.

Answer) Thanks for your comment. At this moment, we apologized that we have no idea about the chemical notation of a-e protons in the substituents because our majors are not chemisty specialist. Therefore, I apologize again I cannot provide chemical notation of a-e proton in the conjugates. Anyway, reaction temperature and reaction time was indicated in the Experimental section.

Figure capton

Figure 1. Synthesis scheme (a) and 1H NMR spectra (b) of chitosan, PEITC and ChitoPEITC conjugates. Chitosan-PEITC conjugation process was carried at 20 oC for 24 hours.

In Experimental section

This mixture was stirred for 24 hours at 20 oC

  1. Figures 1a and 1b are hardly visible. I would recommend increasing their size and resolution.

Answer) Thanks for your comment. According to your comment, we improve the resolution of Figure 1 and enlarge them.

  1. Table 1: Round off the DS values to two significant figures.

Answer) Thanks for your comment. According to your comment, we round off the DS value of CHitoPEITC-2 and CHitoPEITC-3

Table 1. Tables should be placed in the main text near to the first time they are cited.

Feeding ratio (glucosamine:PEITC, mM/mM)

DS of PEITC a

Hydrodynamic radius (nm) b

PDI c

Zeta potential (mV) d

ChitoPEITC-1

ChitoPEITC-2

ChitoPEITC-3

2.0:0.2

2.0:0.4

2.0:0.8

8.6

18

36

89.4±4.3

112.9±5.2

126.2±4.1

0.096

0.116

0.109

12.8±0.53

9.6±0.41

7.1±0.32

a Substitution degree (DS) of PEITC (PEITC number/100 glucosamine) was estimated from 1H NMR spectra in Figure 1. DS of PEITC was calculated as follows: [(Integral value of e proton/2)/(Integral values of C1 proton of chitosan)] × 100.

b Particles sizes were average ± standard deviation from three different measurement.

c Polydispersity index (PDI).

d Zeta potential was average ± standar deviation (S.D.) from three independent measurement.

  1. Table 1: Replace "particle size" with "hydrodynamic diameter" in Table 1 and in the text if these values were determined by dynamic light scattering on the Nano-Zetasizer instrument. Add polydispersity index (PDI) values here and discuss them in the text.

Answer) Thanks for your comment. According to your comment, we corrected them from particle size to hydrodynamic diameter and added PDI values

  1. Tables 1-2 and throughout the text: The standard deviation should be expressed as ONE significant figure; that is, unless the number is between 11 and 19 times some power of ten, in which case you can use two significant figures. The mean value should be rounded off at the decimal place corresponding to the last significant digit of its standard deviation. E.g., 84.4±4.5 (Table 1) should be presented as 84±5, etc.

Answer) Thanks for your comment. According to your comment, we corrected the manuscript in the Tables.

Table 1. Tables should be placed in the main text near to the first time they are cited.

Feeding ratio (glucosamine:PEITC, mM/mM)

DS of PEITC a

Hydrodynamic diameter (nm) b

PDI c

Zeta potential (mV) d

ChitoPEITC-1

ChitoPEITC-2

ChitoPEITC-3

2.0:0.2

2.0:0.4

2.0:0.8

8.6

18

36

89 ± 4

113 ± 5

126 ± 4

0.096

0.116

0.109

13 ± 0.5

9.6 ± 0.4

7.1 ± 0.3

Table 2. Characterization of ChitoPEITC nanophotosensitizers.

ChitoPEITC/Ce6 weight ratio (mg/mg)

Drug contents (%, w/w) a

Loading efficiency (%, w/w) a

Hydrodynamic diameter (nm) b

PDI

Zeta Potential (mV) c

Ce6-ChitooPEITC-1 NP c

40/2

Ce6-ChitoPEITC-2 NP

40/2

40/4

Ce6-ChitoPEITC-3 NP

40/2

4.4

4.5

8.6

4.6

93

95

95

97

133±5

146±4

167±6

173±5

0.114

0.108

0.121

0.113

5.9±0.4

4.2±0.3

2.8±0.4

2.7±0.4

a Drug content (w/w, %) = (Ce6 weight/total weight of nanophotosensitizers) × 100. Loading efficiency (w/w, %) = (Ce6 weight in the nanophotosensitizers/feeding weight of Ce6) × 100.

b Particles sizes were average ± standard deviation from three different measurement.

c Zeta potential was average ± standar deviation (S.D.) from three independent measurement.

d NP means nanophotosensitizers.

  1. Section 2.2: To characterize drug-loaded nanoparticles, data on loading efficiency and encapsulation efficiency are usually provided. I ask you to add this data to the paper.

Answer) Thanks for your comment. According to your comment, we revised the manuscript and then drug contents/loading efficiency was provided.
Table 2. Characterization of ChitoPEITC nanophotosensitizers.

ChitoPEITC/Ce6 weight ratio (mg/mg)

Drug contents (%, w/w) a

Loading efficiency (%, w/w) a

Hydrodynamic diameter (nm) b

PDI

Zeta Potential (mV) c

Ce6-ChitooPEITC-1 NP c

40/2

Ce6-ChitoPEITC-2 NP

40/2

40/4

Ce6-ChitoPEITC-3 NP

40/2

4.4

4.5

8.6

4.6

93

95

95

97

133±5

146±4

167±6

173±5

0.114

0.108

0.121

0.113

5.9±0.4

4.2±0.3

2.8±0.4

2.7±0.4

a Drug content (w/w, %) = (Ce6 weight/total weight of nanophotosensitizers) × 100. Loading efficiency (w/w, %) = (Ce6 weight in the nanophotosensitizers/feeding weight of Ce6) × 100.

b Particles sizes were average ± standard deviation from three different measurement.

c Zeta potential was average ± standar deviation (S.D.) from three independent measurement.

d NP means nanophotosensitizers.

  1. Line 135: I wouldn't call it "burst drug release" because there is almost linear release up to 50 hours.

Answer) Thanks for your comment. According to your comment, we corrected its epression. Practically, drug release pattern was continuous form until 48 h and the release rate was slow down until 96 hours. Then we indicated in the results section.

Ce6 release rate from Ce6-incorporated nanophotosensitizers was almost continuous pattern until 48 hours and then Ce6 was slightly slowly released until 96 hours.

  1. Figure 3: Add the results of the blank experiment (dialysis of pure Ce6) to exclude any artifacts related to dissolution of Ce6 followed by diffusion of Ce6 through the dialysis membrane.

Answer) Thanks for your comment. According to your comments, we performed release experiment of Ce6 and then added to Figure 5.

Figure 5. The effect of polymer (a) and Ce6 contents (b) on the release rate from nanophotosensitizers. These results were average ± standard deviation (SD) from three different experiment.

4.6. Drug release study

The volume of aqueous nanophotosensitizer solution prepared as described above was adjusted to 40 ml (1.0 mg/ml as a ChitoPEITC weight) with deionized water. This solution (5 ml) was introduced into dialysis membrane (MWCO: 2000 Da) and then immersed into 45 ml PBS (pH 7.4, 0.01 M) in conical tube. This was incubated in 37 â—¦C under shaking at 100 rpm. PBS sampled at predetermined time intervals and whole PBS was replaced with fresh PBS. For comparison, pure Ce6 (0.24 mg, similar to Ce6-ChitoPEITC-2 (ChitoPEITC-2:Ce6 = 40:2 weight ratio in Table 2) was dissolved into 5 ml with ultrasonication for 30 s (1 s × 30 times, Vibra-cellTM, Sonics & Materials Inc., Newtown, CT. USA). Then, this solution also introduced into dialysis membrane (MWCO: 2000 Da) and then immersed into 45 ml PBS (pH 7.4, 0.01 M) in conical tube. This was incubated in 37 â—¦C under shaking at 100 rpm. Genesys 10s UV-VIS spectrophotometer (Thermo Fisher Scientific, Waltham, Massachusetts, USA) was employed to measure absorption spectra of this solution. All experiments were carried out at dark condition and the results were expressed as mean ± standard deviation (S.D.) from three different experiments.

  1. Line 295: The chitosan sample must be thoroughly characterized regarding its molecular weight (by viscometry, light scattering, or size exclusion chromatography) and the degree of deacetylation (by NMR, IR, elemental analysis, or titration). The properties of chitosan are very dependent on these parameters; therefore, the wide ranges of values provided by the manufacturer are clearly insufficient. Also indicate the source of chitosan (crab, shrimp, fungi, etc.).

Answer) Thanks for your comment. At this moment, we have some mistake in the writing of chitosan series, i.e. in our laboratory, we have many kinds of chitosan and my students seriously confused the type of chitosan. Practically, we used chitosan oligosaccharide, which was purchased from Tokyo Chemical Industry (TCI) Co., LTD. (Tokyo, Japan). We apologize our mistake. To clarify our mistake, we measured proton and carbon NMR of chitosan oligosaccharide and chitosan 15 k in the supplementary materials as a Figure S1. I confirmed that my students performed all the experiment with chitosan oligosaccharide and other results were identical to chitosan oligosaccharide and then we indicated this statement. 

At this moment, we have no sufficient time to analyze its molecular weight and other charqacteristics (such as deacetylation degree). We added carbon NMR in the Figure S1. Please consider our situation. Thanks to your valuable comments.

  1. Line 327: The authors performed dialysis of loaded nanoparticles for 1 day with frequent water changes. With this methodology, I would have assumed a significant loss of Ce6 drug (about 30% as well as during 24 hours of release, Figure 3), however, according to Table 2, more than 93% of the drug remained encapsulated. How would the authors explain this phenomenon?

Answer) Thanks for your comment. Practically, Ce6 itself is practically insoluble in water while its solubility in PBS (pH 7.4, 0.01M) is higher than deionized water. Because Ce6 has three carboxylic acid and then its solubility is higher in the basic pH. This may be the first reason. Second reason is that, when COOH group of Ce6 complexed with amine group of chitosan in the deionized solution, it is hardly dissociated in the deionized water while it can be slowly dissociated or released from chitosan nanoparticles. Anyway, we discussed more this statement in the discussion section.

In Discussion section

Primarily, cationic polymers such as chitosan or protein is regarded as an ideal candidate for incorporation of Ce6 because of Ce6 has anionic three carboxylic acid [42,43]. Mojzisova et al., reported that Ce6 showed higher solubility in pH 7.4 and its solubility was significantly in the acidic solution [42]. Jeong et al., reported that cationic polymer, chitosan, forms ion-complexes with Ce6 and then formed nanoparticles and then enhances intracellular uptake in vitro [43].

  1. Mojzisova, H.; Bonneau, S.; Vever-Bizet, C.; Brault, D. The pH-dependent distribution of the photosensitizer chlorin e6 among plasma proteins and membranes: a physico-chemical approach. Biophys. Acta. 2007, 1768, 366-374.
  2. Jeong, Y.I.; Cha, B.; Lee, H.L.; Song, Y.H.; Jung, Y.H.; Kwak, T.W.; Choi, C.; Jeong, G.W.; Nah, J.W.; Kang, D.H. Simple nanophotosensitizer fabrication using water-soluble chitosan for photodynamic therapy in gastrointestinal cancer cells. J. Pharm. 2017, 532, 194-203.    

  1. Lines 330-335: It is not clear from this methodology how the complete 100% release of Ce6 from the nanoparticles was achieved? Please clarify.

Answer) Thanks for your comment. First of all, we apologized that my students did some mistake, i.e. they measured Ce6 contents with UV (not fluorescence spectrophotometry). It is practically misunderstand of our students about spectrophotometer. I confirmed that they measured Ce6 contents and release rate with UV spectrophotometer. We calculated that sum of release amount of Ce6 in the release was divided total weight of nanophotosensitizers. Anyway. We indicated it in the experimental section.

In Experimental section

For comparison, 1 mg of Ce6 in DMSO (16 ml) was mixed with 20 mg of empty ChitoPEITC nanoparticles in 4 ml of water following with sonication using ultra-sonicator for 30 s (1 s × 30 times, Vibra-cellTM, Sonics & Materials Inc., Newtown, CT. USA). This solution was diluted with DMSO more than 100 times and then compared with Ce6-incorportaed ChitoPEITC nanophotosensitizers.

Reviewer 3 Report

In this work, a novel chitosan conjugates are reported for the delivery of Ce6 to tumor cells. The fabricated nanoparticles have achieved some meaningful results in photodynamic therapy, which have a beneficial effect on the application of photodynamic therapy. However, there are some problems in the article. The problems are listed as follows:

1. In introduction, “There are few reports regarding to nanoparticles for assistant of ROS production.” This conclusion is not correct, there are many reports regarding to nanoparticles for assistant of ROS production.

2. Are nanoparticles stable under physiological conditions? The size changes of nanoparticles in PBS and in serum-containing medium with time should be suppled.

3.In figure 2, PDI of nanoparticle size should be reported.

4.In figure 2a, the TEM picture is unclear, a clear picture should be provided.

5.In figure 5b, fluorescence pictures need to add scale bar. And it is not certain from the picture that Ce6 is taken up by cells. A clear picture needs to be provided to prove that Ce6 is inside the cells.

6. In Figure 3, the release of Ce6 from nanoparticles appears to be difficult. In the ROS generation experiment and the MTT experiment, the cells were incubated with the nanoparticles for 2 h, was Ce6 released from the nanoparticles at this time? If so, how is it released? If Ce6 works inside nanoparticles without release, what is the significance of Ce6 release experiments from nanoparticles?

Author Response

Response to Reviewer 3’s comment

Thanks for your valuable comments. As far as we can, we revised the manuscript fully.

I

  1. In introduction, “There are few reports regarding to nanoparticles for assistant of ROS production.” This conclusion is not correct, there are many reports regarding to nanoparticles for assistant of ROS production.

Answer) Thanks for your comment. As you commented, we are wrong in this phrases. Then we revised the manuscript and added references. Thanks.

In Introduction

There are many reports regarding to nanoparticles for assistant of ROS production [24-26]. For example, Zhao et al., reported that sonosensitizer -loaded nanoparticles generate ROS and then they can be used for therapeutic or diagnostic purposes [25]. Glass et al., reported that highly tunable nanoparticle-based drug delivery systems improve therapeutic potential of biological agents, i.e. nanoparticles assisted chemotherapy through redox-sensitive drug-release in tumor cells, PDT by enhancing ROS production and radiation therapy by ROS production [26].          Most of the nanoparticulate drug-delivery systems provide delivery platform for anticancer drugs or photosensitizers and then assist ROS-production.

Reference

  1. Yang, B.; Chen, Y.; Shi, J. Reactive oxygen species (ROS)-based nanomedicine. Rev. 2019, 119, 4881-4985. 
  2. Zhao, P.; Deng, Y.; Xiang, G.; Liu, Y. Nanoparticle-assisted sonosensitizers and their biomedical applications. J. Nanomedicine. 2021, 16, 4615-4630.
  3. Glass, S.B.; Gonzalez-Fajardo, L.; Beringhs, A.O.; Lu, X. Redox potential and ROS-mediated nanomedicines for improving cancer therapy. Redox Signal. 2019, 30, 747-761.

  1. Are nanoparticles stable under physiological conditions? The size changes of nanoparticles in PBS and in serum-containing medium with time should be suppled.

Answer) Thanks for your comment. According to your comment, we measured stability of nanoparticles in the deionized water, PBS and RPMI-1640 media. Practically, their stability was properly maintained until 7 days but some of precipitants was observed after 10 days. Anyway, we added these results in the Figure 3 and fully revised.

To analyze stability of particle sizes of nanoparticles, Ce6-incorporated ChitoPEITC nanoparticles (ChitoPEITC NP) in deionized water was mixed with deionized water, phosphate buffered saline and Roswell Park Memorial Institute (RPMI1)-1640 media.    

As shown in Figure 3(a), (b) and (c), average particle sizes were slightly increased according to time course at all aqueous solution. Practically, some of precipitants were observed in the deionized water and PBS of ChitoPEITC-2 and 3 NP sample after 10 days even though they can be easily reconstituted in the media. Even though average particle size was slight increased, stability of ChitoPEITC NP was properly maintained until 7 days in deionized water (Figure 3(a)), PBS (Figure 3(b)) and RPMI-1640 media (Figure 3(c)), Especially, their average particle sizes were slightly higher in the PBS or PRMI-1640 media than those of deionized water.

Figure 4. The stability of particle size of ChitoPEITC NP in deionized water (a), PBS (pH 7.4, 0.01 M) and RPMI1-1640 media (10 % fetal bovine serum, 1 % antibiotics) (c). Aqueous ChitoPEITC NP solution (1 mg/mL in deionized water in Table 2) in deionized water was diluted ten times with deionized water, PBS and RPMI-1640 media. All solutions were stored in the 4 oC until measured them.

3.In figure 2, PDI of nanoparticle size should be reported.

Answer) Thanks for your comment. According to your comment, we indicated the PDI.

4.In figure 2a, the TEM picture is unclear, a clear picture should be provided.

Answer) Thanks for your comment. According to your comment, we changed the TEM picture as far as we can. Thanks again.

Figure 2. Morphological observations using TEM (a) and particle size distribution (b) of ChitoPEITC nanophotosensitizers (C hitoPEITC-2, 20:4 in Table 2).

5.In figure 5b, fluorescence pictures need to add scale bar. And it is not certain from the picture that Ce6 is taken up by cells. A clear picture needs to be provided to prove that Ce6 is inside the cells.

Answer) Thanks for your comment. According to your comment, we observed cell images again and revised the manuscript. Please confirm our results. Thanks again.

  1. In Figure 3, the release of Ce6 from nanoparticles appears to be difficult. In the ROS generation experiment and the MTT experiment, the cells were incubated with the nanoparticles for 2 h, was Ce6 released from the nanoparticles at this time? If so, how is it released? If Ce6 works inside nanoparticles without release, what is the significance of Ce6 release experiments from nanoparticles?

Answer) Thanks for your comment. First of all, Ce6 uptake ratio was higher at ChitoPEITC NP than free ce6 as shown in Figure 6. This result must be induced higher PDT and ROS production. Furthermore, Ce6 release rate and dissociation of nanoparticles must be higher at intracellular environment because endosomal pH is pH 6.5 ~ 5.5. This statement also affects to the results. Anyway, we discussed more in the discussion section.

When nanoparticles were internalized in the cells, they can be partly dissociated due to the acidic pH of endosome and endosomal enzymes following with cellular digestion of the nanoparticles [36]. Due to these properties in the intracellular fate of nanoparticles, drug release must be accelerated in the intracellularly.

  1. Jiang, L.Q.; Wang, T.Y.; Webster, T.J.; Duan, H.J.; Qiu, J.Y.; Zhao, Z.M.; Yin, X.X.; Zheng, C.L. Intracellular disposition of chitosan nanoparticles in macrophages: intracellular uptake, exocytosis, and intercellular transport. J. Nanomedicine. 2017, 12, 6383-6398. 

Round 2

Reviewer 2 Report

The authors did a great job revising the manuscript, and I am very appreciative of that. However, there are still a couple of points for authors to consider:

Major point:

To my regret, I cannot accept the authors' reasoning for the complete failure to characterize chitosan (chitosan oligosaccharide). Most of us involved in "chitin science" use commercial samples of chitosan, but we know that the characteristics (molecular weight, dispersity, degree of deacetylation) of these commercial samples vary greatly from batch to batch. Numerous papers on chitosan show that the properties of this polysaccharide are very dependent on its characteristics, so the European Chitin Society, which is affiliated with IJMS, does not recommend the publication of results on chitosan without explicitely indicating its characteristics. Otherwise, the evaluation of any properties of chitosan becomes meaningless.

At least three characteristics for chitosan must be specified or determined with sufficient accuracy:

1) source of chitosan (crab, shrimp, fungi, etc.). 

2) molecular weight (by viscometry, light scattering, or size exclusion chromatography) 

3) degree of deacetylation can be calculated from the 1H NMR spectrum of chitosan shown on Figure 1b.

Minor point:

Figure 1a: Draw the structures of chitosan and its derivative, avoiding the use of angles around the glycosidic bond (the angle in the structural formula denotes a carbon atom). The correct structural formula for chitosan can be found, for example, in Wikipedia https://en.wikipedia.org/wiki/Chitosan

Author Response

Major point:

To my regret, I cannot accept the authors' reasoning for the complete failure to characterize chitosan (chitosan oligosaccharide). Most of us involved in "chitin science" use commercial samples of chitosan, but we know that the characteristics (molecular weight, dispersity, degree of deacetylation) of these commercial samples vary greatly from batch to batch. Numerous papers on chitosan show that the properties of this polysaccharide are very dependent on its characteristics, so the European Chitin Society, which is affiliated with IJMS, does not recommend the publication of results on chitosan without explicitely indicating its characteristics. Otherwise, the evaluation of any properties of chitosan becomes meaningless.

At least three characteristics for chitosan must be specified or determined with sufficient accuracy:

1) source of chitosan (crab, shrimp, fungi, etc.). 

Answer) Thanks for your comment. We can get a information about origin of COS from manufacturer. Origin of COS was crustacea. We indicated it in the Experimental section.

In Experimental section

From manufacturer’s data, origin of COS was crustacea and deacetylation degree was approximately 94 % (Figure 1(b) and Figure S2). Molecular weight was calculated as 1150 g/mol (Figure S3) from quantitative NMR (qNMR) spectra.

2) molecular weight (by viscometry, light scattering, or size exclusion chromatography) 

Answer) Thanks for your comment. To analyze molecular weight of COS with Gel permeation chromatography, we have tried to find GPC equipment (which is operated based on water) in our city and state for two weeks. Unfortunately, we couldn’t find GPS for water as a mobile phase. Some research center has a GPC, which is operated with THF as a mobile phase. We told them, we will provide column for use of water as mobile phase. However, they told us their GPC already reserved until end of December and our sample can be analyzed at January of 2023.

Then, we analyzed quantitative NMR using dimethylmalonic acid as a standard material. We obtained a result and calculated M.W. of COS was 1,150 g/mol. This result was added to Figure S3 Furthermore, we also added MALDI-TOF MS results in the Figure S4. MALDI-TOF MS results indicated that glucosamine number of COS was variable between 4 and 11. Anyway, we have some difficulties in measurement of GPC and then we added qNMR and MALDI-TOF data. Please consider our efforts to address your comment. We appreciated again.

In Experimental section

From manufacturer’s data, origin of COS was crustacea and deacetylation degree was approximately 94 % (Figure 1(b) and Figure S2). Molecular weight was calculated as 1150 g/mol (Figure S3) from quantitative NMR (qNMR) spectra.

In Results section

1H and 13C nuclear magnetic resonance (NMR) spectra of COS shows specific peaks of H1 ~ H7 as shown in Figure S1(a) and (b). That is, acetyl group of COS was observed at 1.85 ppm and H3 ~ H6 was observed at 3.4 ~ 3.9 ppm. H1 and H2 was observed at 4.6 ~ 4.8 ppm and 2.7 ~ 3.0 ppm. Furthermore, 13C NMR of COS also showed specific peaks of each position of carbon as shown in Figure S1(b). To calculate deacetylation degree of COS, integral value between H3 ~ H6 and H7 was compared and calculated deacetylation degree was approximately 96 % as shown in Figure S2. Figure S3 was quantitative NMR (qNMR) of COS for evaluation of molecular weight (M.W.) of COS. For evaluation of M.W. of COS, dimethylmalonic acid was used as a standard material. Integral value of H2 and methyl proton of dimethylmalonic acid was compared and then typical M.W. of COS was calculated as 1150 g/mol. Furthermore, analysis of COS using matrix-assisted laser desorption ionization mass (MALDI TOF/TOF MS) spectrometer showed that COS was composed of various M.W. of oligosaccharide existed as shown in Figure S4.

In Supplement material

Figure S3. qNMR spectra of COS for evaluation of MW. For evaluation of COS M.W., dimethylmalonic acid was used as a standard material. Th specific peaks of dimethylmalonic acid was observed in 1.23 ppm. Based on integral value in NMR and M.W. of dimethylmalonic acid, MW of COS was calculated and then M.W. to 1,832 g/mol. COS and dimethylmalonic acid were dissolved in 1 ml DMSO/D2O mixtures (1/1, v/v). M.W. of COS was calculated with following equation:

I : integral area

N : number of nuclei

C : concentration of the compound of interest (x), calibrant (cal)

Figure S4. MALDI TOF/TOF MS spectrum of COS.

3) degree of deacetylation can be calculated from the 1H NMR spectrum of chitosan shown on Figure 1b.

Answer) Thanks for your comment. According to your comment, we calculated the deacetylation degree was calculated and indicated in the Figure 1b, Figure S2 and Experimental section.

In Experimental section

From manufacturer’s data, origin of COS was crustacea and deacetylation degree was approximately 94 % (Figure 1(b) and Figure S2). Molecular weight was calculated as 1150 g/mol (Figure S3) from quantitative NMR (qNMR) spectra.

Figure S2. 1H NMR spectra of COS for evaluation of deacetylation degree of COS. To calculate deacetylation degree of COS, integral value of (H3 ~ H6) and H7 was compared and then value of deacetylation degree was calculated as 96.13.

Minor point:

Figure 1a: Draw the structures of chitosan and its derivative, avoiding the use of angles around the glycosidic bond (the angle in the structural formula denotes a carbon atom). The correct structural formula for chitosan can be found, for example, in Wikipedia https://en.wikipedia.org/wiki/Chitosan

Answer) Thanks. I appreciated your comment. According to your comment, corrected the molecular formular based on Wikipedia https://en.wikipedia.org/wiki/Chitosan. I appreciated your comment because we learned many things from your comment.

Reviewer 3 Report

The author has carefully replied all questions. This manuscript can be published without further revision.

Author Response

The author has carefully replied all questions. This manuscript can be published without further revision.

Answer) Thanks to your comments. I appreciated your positive comments. Thanks.

Round 3

Reviewer 2 Report

I am grateful to the authors for their efforts to improve the paper.